# Regulation of chromatin microphase separation by binding of protein complexes

**Omar Adame-Arana[1]\*, Gaurav Bajpai[1], Dana Lorber[2], Talila Volk[2], Samuel Safran[1]**

[1]Department of Chemical and Biological Physics, Weizmann Institute of Science, Rehovot, Israel; [2]Department of Molecular Genetics, Weizmann Institute of Science, Rehovot, Israel

**Abstract** We show evidence of the association of RNA polymerase II (RNAP) with chromatin in a core-shell organization, reminiscent of microphase separation where the cores comprise dense chromatin and the shell, RNAP and chromatin with low density. These observations motivate our physical model for the regulation of core-shell chromatin organization. Here, we model chromatin as a multi-block copolymer, comprising active and inactive regions (blocks) that are both in poor solvent and tend to be condensed in the absence of binding proteins. However, we show that the solvent quality for the active regions of chromatin can be regulated by the binding of protein complexes (e.g., RNAP and transcription factors). Using the theory of polymer brushes, we find that such binding leads to swelling of the active chromatin regions which in turn modifies the spatial organization of the inactive regions. In addition, we use simulations to study spherical chromatin micelles, whose cores comprise inactive regions and shells comprise active regions and bound protein complexes. In spherical micelles the swelling increases the number of inactive cores and controls their size. Thus, genetic modifications affecting the binding strength of chromatin-binding protein complexes may modulate the solvent quality experienced by chromatin and regulate the physical organization of the genome.

**\*For correspondence:** omar.adame-arana@weizmann. ac.il

**Competing interest:** The authors declare that no competing interests exist.

## Editor's evaluation

This fundamental work substantially advances our understanding of polymer physics underpinnings of genome folding, organization, and regulation. The conclusions are supported by both convincing computer simulations and analytical theory. The work will be of significant interest to the genome folding community.

## Introduction

The spatiotemporal organization of the genome is of utmost importance for the normal functioning of cells. The genome of eukaryotic cells is packaged in the chromatin fiber, which is the biomolecular complex made of DNA wound around histone proteins (*Kornberg and Thomas, 1974*; *Kornberg, 1974*; *Luger et al., 1997*). The chromatin fiber is a long chain of nucleosomes connected by sections of linker DNA with a typical size of 20–60 base pairs. At low salt concentrations, in vitro experiments characterize chromatin by a beads-on-a-spring architecture with a width of 10 nm (*Olins and Olins, 1974*; *Oudet et al., 1975*; *Woodcock et al., 1976*). In solutions with a higher salt content, a 30 nm more compact fiber of chromatin is measured (*Oudet et al., 1975*). However, 30 nm fibers have not been found in living cells (*van Holde and Zlatanova, 1995*; *Eltsov et al., 2008*; *Nishino et al., 2012*; *Bajpai et al., 2017*). Instead, chromatin organizes in domains of different sizes that seem to be formed

by many interdigitated fibers of chromatin of different sizes (*Maeshima et al., 2014*; *Ou et al., 2017*; *Bajpai and Padinhateeri, 2020*). This suggests several different mechanisms of chromatin compaction, which has been examined in more detail in in vitro reconstitution of short arrays of chromatin (*Maeshima et al., 2016b*). There it was established that these short strands of chromatin self-associate to form globular structures. Although the liquid-like behavior of chromatin was described for chromatin in vivo (*Maeshima et al., 2016a*), the material properties of such chromatin globular structures which behave like liquid drops was not established at that time.

More recently, however, in vitro experiments have shown that the globular chromatin structures arising from self-association are related to an intrinsic tendency of chromatin to form liquid-like phases (*Gibson et al., 2019*). This phase separation can be disrupted by acetylation of histone tails, but phase separation is recovered with the addition of a protein with multiple bromodomains. These liquid-like drops composed of acetylated chromatin and bromodomains coexist and associate with the phase-separated, non-modified chromatin droplets, suggesting that liquid-liquid phase separation may play a role on the regulation of genome organization. Further evidence of the intrinsic ability of chromatin to phase separate in vitro has been given (*Strickfaden et al., 2020*), although these authors reached different conclusions regarding the material properties of such condensed phases. In addition, there is a growing body of in vivo evidence, of phase separation of chromatin from the nucleoplasm (*Popken et al., 2014*; *Strickfaden et al., 2020*; *Amiad-Pavlov et al., 2021*; *Belew et al., 2021*). In particular, it has been shown in several different cell types that chromatin organizes preferentially at the nuclear periphery and is depleted in the central region of the nucleus (*Bhattacharya et al., 2009*; *Popken et al., 2014*; *Amiad-Pavlov et al., 2021*; *Bajpai et al., 2021*; *Belew et al., 2021*). In *Amiad-Pavlov et al., 2021*, it was further shown that in intact, live *Drosophila* cells, both acetylated chromatin (commonly associated with active genes in euchromatin) and inactive chromatin (commonly associated with heterochromatin) are preferentially found in the periphery of the nucleus and depleted in the central region. Furthermore, at the nuclear periphery they observed a spatial pattern in which regions of chromatin with H3K9ac modifications and regions of high chromatin density were further demixed. Thus both types of chromatin tend to demix from the aqueous phase and then from each other. The experiments also show chromatin regions in which there are no H3K9ac modifications positioned near the lamina in an organization similar to a wetting droplet. In this paper, we address how the organization of active and inactive regions of chromatin can be regulated by the reversible binding of protein complexes. Hereafter, we use the name active for regions of chromatin which are found in regions of lower chromatin density that would commonly be associated with genes that may undergo transcription, It is important to note that this use of active is different from activity as used in the physics literature (e.g., active swimmers) on non-equilibrium forces in systems with internal energy sources.

The observed patterns described above are reminiscent of microphase separation of block copolymers, in which two phases, each enriched in one or the other type of blocks, separate from each other. In contrast to a system undergoing macrophase separation, the microphases do not fully coarsen due to the chain connectivity and the length scales defining the spatial patterns are determined by the size of the blocks and their intrinsic properties, such as self-attraction and bending rigidity (*de Gennes, 1978*; *Leibler, 1980*; *Semenov, 1985*; *Ohta and Kawasaki, 1986*; *Olvera de la Cruz, 1989*; *Halperin, 1991*). Chromatin microphase separation seems to be a natural consequence of the distribution of active and inactive regions within a chromosome that are similar to a long, multiblock copolymer. Such regions may correlate with epigenetic modifications (e.g., H3K9ac, as mentioned above) (*Jost et al., 2014*) and different guanine-cytosine (GC) content (*Dekker, 2007*; *Jabbari and Bernardi, 2017*). The fact that there are chromatin regions (or blocks) with different biochemical properties, naturally raises the possibility of microphase separation, which might be responsible for the compartmentalization of chromatin into so-called A and B compartments (*Lieberman-Aiden et al., 2009*). Indeed, this idea has been put forth by a number of authors where microphase separation has been invoked to explain the formation of such compartments (*Jost et al., 2014*; *Sazer and Schiessel, 2018*; *Nuebler et al., 2018*; *Brackley and Marenduzzo, 2020*; *Belaghzal et al., 2021*; *Sommer et al., 2022*). The differences in chromatin density in the compartments might dictate whether the transcription machinery can interact with and activate genes. Such physical mechanism was considered in describing how small transcription complexes binding to active genes can be used to keep these genes at the surface of more condensed chromatin domains (*Maeshima et al., 2015*). Interestingly, recent high-resolution

microscopy has revealed that the transport of nano-particles of different sizes strongly depends on the chromatin density (degree of condensation), and demonstrate that active chromatin regions are more accessible as compared to inactive regions (*Gelléri et al., 2022*).

Several papers called attention to the fact that chromatin might organize in a microphase-separated state with strings of spherical micelles (*Ostashevsky, 2000*; *Ostashevsky, 1998*; *Sommer et al., 2022*). One model proposed that micelle cores comprise chromatin regions with high GC content (associated with genes that are more active) that can be cross-linked by multi-protein complexes while the micelle shells contain AT-rich chromatin regions (associated with inactive genes) (*Ostashevsky, 2000*; *Ostashevsky, 1998*). Although the aforementioned model is questionable in light of experimental observations that show that inactive regions of chromatin, such as heterochromatin, form spherical, phase-separated domains (*Strom et al., 2017*), it correctly points to the fact that chromatin will very likely form micellar structures due to its block copolymer nature. Moreover, despite the relevance of microphase separation to compartmentalization of chromatin, there is not yet an intuitive and minimal analytical model of the regulation of such microphase separation. A previous theoretical paper focused on the effect of RNA Pol II, in chromatin organization by considering chromatin polymer brushes in which transcription can lead to *lateral* phase separation of nucleosome-rich and nucleosome-poor regions (*Yamamoto and Schiessel, 2016*). Another recent article showed that transcription seems to organize euchromatin as a microphase-separated state, with RNA Pol II acting like an amphiphile that connects a DNA-rich phase with an RNA-rich phase (*Hilbert et al., 2021*). The authors performed Monte Carlo simulations to explain the experimentally observed microphase separation. In subsequent studies (*Pancholi et al., 2021*; *Hajiabadi et al., 2022*), it was shown that RNA Pol II exhibit different types of organization depending on its phosphorylation state and interactions with active genes. Our findings complement these previous studies by showing experimental evidence of a robust core-shell organization of chromatin and RNA Pol II. These observations are made in intact, live organisms (*Drosophila* larvae muscle fibers), which exhibit peripheral chromatin organization and additionally exhibit microphase separation since the chromatin domains do not fully coarsen (*Amiad-Pavlov et al., 2021*).

Based on our new observations we propose a minimal model of chromatin in which the microphase separation between active and inactive regions is regulated by the reversible binding of protein complexes to active chromatin regions. Our model is based on the observed self-association of chromatin (in the absence of chromatin-binding protein complexes) and shows that the relatively uncondensed nature of active chromatin can be regulated by reversible binding of protein complexes, resulting in an inner core of condensed (inactive) chromatin connected to a corona of relatively uncondensed (active) chromatin to which protein complexes are bound. We predict the relative sizes and organizations of these regions into a structure that contains an assembly of such core-shell micelles. Those are all connected and thus localized in space since they arise from the block copolymer-like structure of the chromatin with regions of acetylated and non-acetylated (e.g., unmodified and methylated) chromatin. Interestingly, the binding of protein complexes can lead to changes in the organization of both, active and inactive chromatin regions, despite protein complexes interacting only with the active regions.

Our manuscript is organized as follows. We first show observations of a core-shell organization of chromatin and RNA Pol II. Inspired by these observations, we introduce our minimal model of regulation of microphase separation by the binding of protein complexes and discuss the equilibrium conditions in the system. We then discuss analytical and simulation results, which show that the binding of protein complexes to chromatin can regulate the solvent quality for the active chromatin regions. We conclude by discussing our findings in the context of chromatin organization in the nucleus and briefly comment on an extension to non-equilibrium binding of protein complexes to chromatin.

## Results
### Spatial organization of chromatin and RNA Pol II in intact, live *Drosophila* cells

Chromatin in intact, live tissue *Drosophila* cells organizes peripherally, with a chromatin-depleted central region in the nucleus (*Amiad-Pavlov et al., 2021*). Such organization differs from the conventional picture of chromatin as a polymer network that fills the entire nucleus (*Bickmore, 2013*; *Misteli,*

2020; *Kempfer and Pombo, 2020*) and motivates further investigation of the smaller scale structure of the peripheral chromatin layer. Experimental evidence shows that despite the tendency of chromatin to phase separate from the nucleoplasm, it does not behave like a simple liquid that wets the entire nuclear lamina. Instead, it further assembles into a series of dense chromatin domains distributed along the periphery (*Amiad-Pavlov et al., 2021*). The physical mechanisms leading to such a distribution of dense chromatin cores parallel to the nuclear envelope remain elusive. Some hints come from evidence showing that chromatin regions with H3K9ac modification, a marker of active chromatin, are also found along the periphery, but they are distributed around the dense chromatin domains. This seems to be a microphase-separated state, in which dense chromatin occupies the core of such domains and chromatin with H3K9ac modifications is only found at the shell, such that the chromatin domains exhibit a core-shell organization. If indeed active marks are related to transcription, where is the transcription machinery located? In particular, what is its role in shaping genome organization?

Live imaging of intact, live *Drosophila* third instar larvae in which total chromatin, labeled with His2B-mRFP expressed by its own promoter, and RNA Pol II complex, visualized by muscle-specific expression of Rpb3-GFP driven by the Mef2-GAL4 driver, was performed by the following procedure:

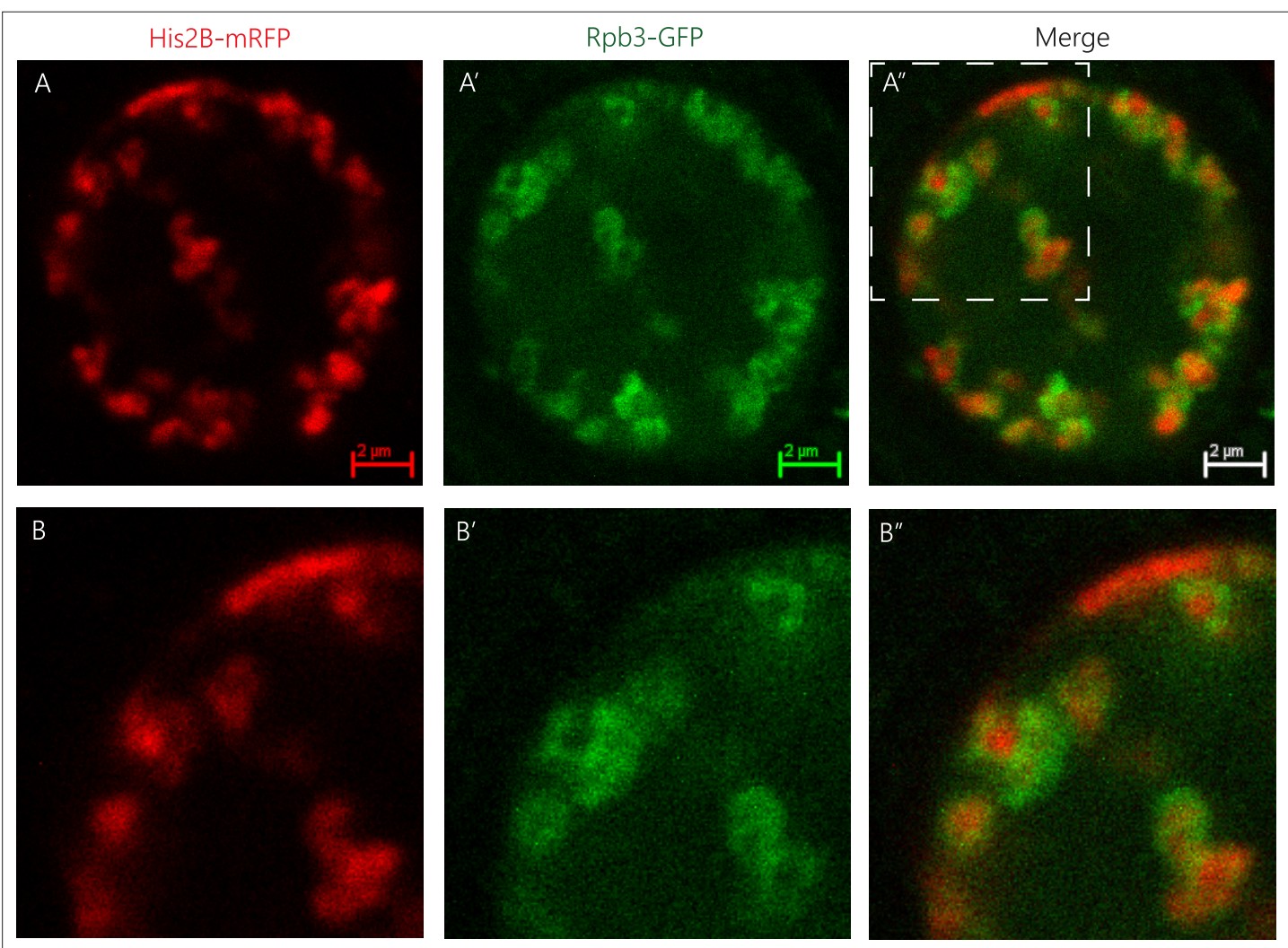

**Figure 1.** Spatial organization of RNA Pol II and chromatin. Representative image of a muscle nucleus of live *Drosophila* third instar larvae labeled with Histone2B-mRFP (**A**, red, expressed under endogenous promoter) and with Rpb3-GFP (**A′**, green, expressed in muscles under Mef2-GAL4 driver), and their merged images (**A″**). **B–B″** panels are corresponding enlargements of the rectangle area illustrated in **A″**. Note that the RNA Pol II green signal (Rpb3-GFP), representing transcribed regions, is observed mostly at the circumference of clusters of non-transcribed DNA depicted by the histone2B-associated red signal.

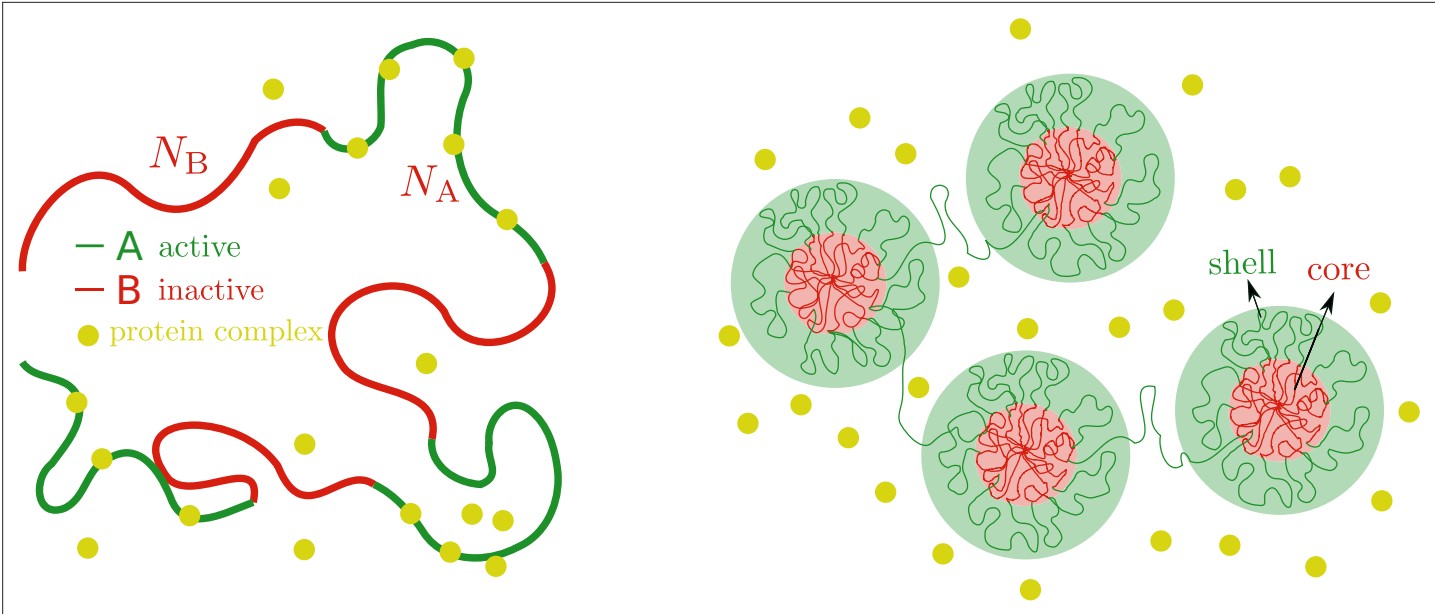

**Figure 2.** Microphase separation of active and inactive chromatin regions. Left: Chromatin as a multiblock copolymer with active (green) and inactive blocks (red) comprising $N_A$ and $N_B$ monomers, respectively. Right: Core-shell organization of chromatin. The cores (red area) are composed of condensed, inactive blocks of chromatin and the shells (green area) are composed of active blocks of chromatin that are less condensed that the cores in the absence of chromatin-binding protein complexes that cause them to be even more swollen.

Live larvae with both fluorescent marks were inserted in a special device (*Lorber et al., 2020*), and visualized by high-resolution imaging using an inverted Leica SP8 STED3A microscope as described in *Amiad-Pavlov et al., 2021*. A representative single confocal image (see *Figure 1*) shows that RNA Pol II is found distributed around dense chromatin domains. In Appendix 1 we show another representative confocal image of the distribution of chromatin and RNA Pol II in a salivary gland of an intact *Drosophila* larva. This novel observation and previous experiments in zebrafish embryos in the late blastula stage (*Hilbert et al., 2021*) motivate our theory of a minimal model of chromatin organization in which we consider different regions of chromatin that are microphase separated and that can additionally be regulated by the reversible binding of protein complexes, such as RNA Pol II but more generally, by any of the proteins that interact with active regions of chromatin.

## Theoretical model: chromatin as a multiblock copolymer

We model chromatin as a multiblock copolymer with two types of blocks, active (A) and inactive (B) blocks, see *Figure 2A*. The A blocks represent active chromatin regions (active genes) and the B blocks inactive chromatin regions (silent genes), which could roughly correspond to euchromatin and facultative heterochromatin, respectively (*Lieberman-Aiden et al., 2009*; *Hildebrand and Dekker, 2020*; *Mirny and Dekker, 2022*). For simplicity, we treat a chromosome as a polymer made of $N$ total monomers with a periodic pattern of $N_A$ monomers of type A, followed by $N_B$ monomers of type B. The total number of monomers is given by $N = n(N_A + N_B)$, where $n$ is the number of blocks of each type in the polymer. For simplicity we consider the case where both types of monomers have the same molecular size $a$. Motivated by recent experimental and theoretical studies on chromatin organization showing that chromatin behaves as a self-attractive polymer (polymer in poor solvent) (*Amiad-Pavlov et al., 2021*; *Bajpai et al., 2021*; *Adame-Arana and Safran, 2021*), we model both blocks (in the absence of chromatin-binding protein complexes) as being in poor solvent conditions; both blocks phase separate from the aqueous phase, consistent with the experiments. However, due to the difference in attraction of AA vs. BB, the two types of blocks will undergo a further, microphase separation, reminiscent of the data in *Amiad-Pavlov et al., 2021*, where acetylated chromatin is mostly found surrounding regions of higher density of chromatin. This is in agreement with the well-established fact that chromatin separates into different compartments (*Lieberman-Aiden et al., 2009*; *Mirny et al., 2019*; *Hildebrand and Dekker, 2020*). In general, microphase separation can take different forms,

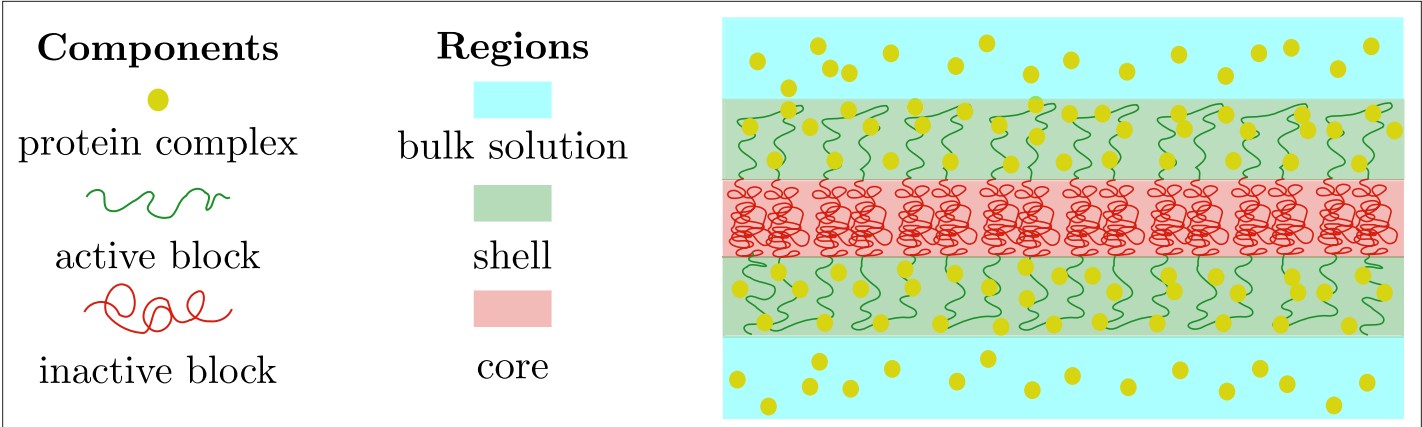

**Figure 3.** Microphase separation of chromatin in a layer geometry. The microphase-separated state in the layer geometry is characterized by three regions: The core (red shaded area), which is composed of inactive blocks (red curves) and solvent (not shown), the shell (green shaded area), comprising active blocks (green curves), free and bound protein complexes (yellow circles) and solvent (not shown), and finally the bulk solution (blue shaded area) of protein complexes (yellow circles).

where the selectivity of the solvent (poorer solvent for the inactive blocks) leads to lamellar, spherical, or cylindrical organization depending on the length of the different blocks, the type of solvent, and the interactions between the blocks (*de Gennes, 1978*; *Semenov, 1985*; *Ohta and Kawasaki, 1986*; *Olvera de la Cruz, 1989*; *Zhulina et al., 2005*). In *Figure 2B*, as an example, we show a string of micelles where the core is made of strongly attractive inactive blocks and the shell is composed of active blocks. In what follows we describe a minimal model of the regulation of chromatin microphase separation where the inactive blocks form a dense core (inactive core) and the active blocks are in a more dilute shell (active shell) which can be accessed by protein complexes that reversibly bind to the active blocks. Our model considers microphase separation in the strong segregation limit (*Semenov, 1985*; *Ohta and Kawasaki, 1986*) with sharp interfaces, the strong incompatibility arises here from the strong selectivity of the solvent, in which inactive blocks experience very strong compaction, reminiscent of the much higher density of DNA in inactive compartments. Hereafter, we omit the word reversibly and binding should be understood in the remaining of the text as equilibrium reversible binding unless otherwise stated. Our assumption that protein complexes only bind to active monomers stems from the observed lack of penetration of RNA Pol II into the dense chromatin cores as shown in *Figure 1*. Additionally, there is further evidence showing that large particles cannot diffuse into inactive chromatin domains (*Gelléri et al., 2022*).

## Core-shell organization of chromatin in a solution of protein complexes

In order to simplify our discussion, we focus on the layer geometry shown in *Figure 3* and later on apply our results to spherical domains. The system has three different regions. Two of these regions are the core, which contains the inactive blocks and some solvent and the shell which contains the active blocks, solvent, and protein complexes; these can either be bound to the active blocks or free in solution in the volume defined by the extension of the active blocks. Finally, there is the bulk solution, in which there are only protein complexes and solvent, which acts as a reservoir of protein complexes for the core-shell system. Below, we define the free energy contributions of each of these regions.

### Inactive core

The core composed of inactive blocks and some solvent has an interfacial area, $S$, at each of its interfaces with the shell and a thickness $2L_B$. The volume fraction of the inactive blocks within the core is $\phi_B$ and in the limit of attractive interactions among the inactive blocks that are much larger than those of the active blocks or the binding energy of the protein to the active blocks, this volume fraction is a constant. Such an approximation is motivated by the fact that dense chromatin domains are known to have much higher concentrations that chromatin found in transcriptionally active compartments (*Gelléri et al., 2022*). We now define the free energy of the core as

$$F_{\text{core}} = 2\gamma S, \tag{1}$$

and the conservation of B monomers by

$$a^3 n N_{\text{B}} = 2\phi_{\text{B}} L_{\text{B}} S. \tag{2}$$

In the expression for the free energy of the core, *Equation 1*, we did not consider the elastic contribution from the inactive blocks because it is usually negligible compared to the interfacial tension and other contributions (*Zhulina et al., 2005*).

## Active shell

The spatial extent of the active shell region is determined by the extension of the loops of active blocks, $L_{\text{A}}$, which emanate from the core. The volume of the active shell region is given by $V_{\text{shell}} = 2L_{\text{A}}S$. The composition variables in the shell are: the volume fraction of the active monomers, $\phi_{\text{A}}$, the probability that an active monomer is bound to a protein complex, $\Omega$, and the volume fraction of free protein complexes within the solvent of the shell, $\phi_{\text{fc}} = a^3 N_{\text{fc}}/(2L_{\text{A}}S)$. Here, we have introduced the total number of free protein complexes in the shell region, $N_{\text{fc}}$. We now define the free energy of the shell, $F_{\text{shell}}$, as the sum of three contributions,

$$F_{\text{shell}} = F_{\text{mix}} + F_{\text{bin}} + F_{\text{brush}}. \tag{3}$$

The first contribution is the mixing entropy of the solvent molecules and the protein complexes, which is given by

$$F_{\text{mix}} = \frac{2L_{\text{A}}S k_{\text{B}}T}{a^3} \Big( \phi_{\text{fc}} \log \phi_{\text{fc}} + \big(1 - \phi_{\text{fc}} - \phi_{\text{A}} \\ -\phi_{\text{A}}\Omega \log(1 - \phi_{\text{fc}} - \phi_{\text{A}} - \phi_{\text{A}}\Omega)\Big). \tag{4}$$

The second contribution is the decrease in free energy due to the binding of protein complexes to the active blocks

$$F_{\text{bind}} = n N_{\text{A}} k_{\text{B}}T \left( \Omega \log \Omega + (1 - \Omega)\log(1 - \Omega) - \epsilon\Omega \right),$$

here, the first and second terms are entropic contributions associated with the number of available active monomers to which a protein complex can bind, and the third term is the energy gain, $\epsilon$, of a protein complex that binds to an active block. The last contribution to the free energy of the shell region, $F_{\text{brush}}$, stems from the active blocks and we choose it to be identical with that of a polymer brush. This free energy includes the polymer stretching energy and the interactions between the monomers of the active blocks and is given by

$$F_{\text{brush}} = n k_{\text{B}}T \left( \frac{3L_{\text{A}}^2}{N_{\text{A}}a^2} + \frac{v N_{\text{A}}\phi_{\text{A}}(1 - \Omega)^2}{2} \right). \tag{5}$$

where we introduced the interaction between active monomers, $v$, which we consider to be attractive (in the absence of protein binding), that is $v < 0$. One core assumption in our model is that the presence of bound protein complexes reduces the effective attraction between active monomers. The rationale is based in the following: The protein complexes that bind to active chromatin tend to be relatively large (e.g., RNA Pol II pre-initiation complex), in which a myriad of proteins assemble jointly with RNA Pol II in order to initiate transcription, which can lead to steric repulsion due to the large macromolecular volume of such complexes (*Hahn, 2004*). Moreover, while RNA Pol II interacts with the DNA in the chromatin fiber via positively charged patches at its surface, most of its surface is negatively charged (*Cramer et al., 2001*). Hence, we assume that when RNA Pol II is bound to the chromatin fiber, those Pol II regions that are negatively charged would effectively provide a repulsive force within the chromatin fibers. Another possible explanation is that in the vicinity of active monomers, the transcription complex carries mRNA, which is known to segregate from chromatin (*Hilbert et al., 2021*). While a more thorough analysis of the molecular interactions would be interesting, it is out of the scope of our paper that uses a coarse-grained, polymer physics approach. This approach

also allows our model to be predictive as to the physical organization and growth of the domains, independent of those molecular details that are as yet unknown.

The free energy we consider in *Equation 5* is appropriate in the limit that the surface density of the contacts between active and inactive blocks always remains relatively high; therefore, active blocks are heavily influenced by the presence of neighboring blocks. In this limit, the system behaves like a polymer brush. In the other limit that the surface density of contacts of the active and inactive brushes is sufficiently low, the appropriate expression would be that of isolated mushrooms in which each block is independent of each other and their description is equivalent to that of single polymers which obey simple scaling laws, depending on the solvent quality (*Szleifer, 1996*).

The conservation of active monomers in the shell is expressed by

$$a^3 n N_a = 2\phi_A L_A S.$$ (6)

In writing the free energy of the brush as in *Equation 5*, we made the simplifying assumption that the contribution of each loop of active blocks is the sum of the contributions of the two independent halves of a loop, disregarding the fact that the loops are closed. Since the closure only affects the conformations of the monomers at the ends of each brush, this approximation is appropriate when the brushes contain many monomers.

## Bulk solution

The bulk solution of protein complexes occupies a volume $V_{bulk}$ whose free energy in the dilute limit is

$$F_{bulk} \approx \frac{k_B T V_{bulk}}{a^3} \left( \phi_{bulk} \log \phi_{bulk} - \phi_{bulk} \right),$$ (7)

where $\phi_{bulk}$ is the volume fraction of protein complexes in the bulk. Moreover, we consider the bulk solution to be much larger than the core and shell regions, so that it acts effectively as a reservoir of protein complexes whose exchange chemical potential is

$$\mu_{bulk} = \left( \frac{\partial F_{bulk}}{\partial N_{bulk}} \right) \Big|_{V_{bulk}} = k_B \log \phi_{bulk},$$ (8)

The osmotic pressure of the bulk solution is

$$\Pi_{bulk} = - \left( \frac{\partial F_{bulk}}{\partial V_{bulk}} \right) \Big|_{N_{bulk}} = \frac{k_B T \phi_{bulk}}{a^3},$$ (9)

where we used the relation $\phi_{bulk} = a^3 N_{bulk}/V_{bulk}$. In equilibrium, the exchange chemical potentials of the protein complexes must be equal in the coexisting bulk and core-shell regions and the osmotic pressures must be balanced.

## Equilibrium conditions for the chromatin core-shell system

The total energy of the system, $F$, is then given by

$$F = F_{core} + F_{shell} + F_{bulk},$$ (10)

where $F_{core}$, $F_{shell}$, and $F_{bulk}$ are the free energy of the core, shell, and bulk solution, previously defined in *Equations 1, 3, and 7*, respectively. The degrees of freedom to be determined are: the thickness of the shell, $L_A$, the interfacial area, $S$, and the number of free and bound protein complexes, $N_{fc}$ and $N_{bc} = n\Omega N_A$, respectively. The thermodynamic conditions for coexistence of the bulk solution with the core-shell structure are given by requiring equality of the exchange chemical potentials of the protein complexes within the shell (bound and free) and the protein complexes in the bulk solution as well as the balance of osmotic pressures (see Appendix 2 for details).

The equalities of the exchange chemical potentials of protein complexes in the system can be expressed as

$$\mu_{fc} = \mu_{bc},$$ (11)

$$\mu_{fc} = \mu_{bulk},$$ (12)

where the exchange chemical potential of the free protein complexes is

$$\mu_{\text{fc}} = \left( \frac{\partial F_{\text{shell}}}{\partial N_{\text{fc}}} \right)_{L_A, S, N_{\text{bc}}},\tag{13}$$

and the exchange chemical potential of the bound protein complexes is given by

$$\mu_{\text{bc}} = \left( \frac{\partial F_{\text{shell}}}{\partial N_{\text{bc}}} \right)_{L_A, S, N_{\text{fc}}}.\tag{14}$$

The third condition is the equality of the osmotic pressures in the shell region and bulk solution

$$\Pi_{\text{shell}} = \Pi_{\text{bulk}},\tag{15}$$

where the osmotic pressure in the shell, $\Pi_{\text{shell}}$, due to a change in the brush length $L_A$ is given by

$$\Pi_{\text{shell}} = -\frac{1}{2S} \left( \frac{\partial F_{\text{shell}}}{\partial L_A} \right)\Big|_{S, N_{\text{bc}}, N_{\text{fc}}}.\tag{16}$$

The free energy in the core-shell region also varies with changes in the interfacial area $S$, so the osmotic pressure equality also implies:

$$\left( \frac{\partial F}{\partial S} \right)\Big|_{L_A, N_{\text{bc}}, N_{\text{fc}}} + 2\Pi_{\text{shell}} L_A = 0.\tag{17}$$

The equilibrium properties of the system are then found by simultaneously solving *Equation 11*, *Equation 12*, *Equation 15*, and *Equation 17*.

## Protein complexes regulate the extension of the active chromatin blocks

We first focus on the effect that protein complexes have on the stretching of the polymer brush made of active chromatin blocks; hereafter, we refer to such polymer brush as the active chromatin brush. In order to find analytical solutions, we work in the limit of dilute protein complexes in (i) the bulk $\phi_{\text{bulk}} \ll 1$ and (ii) the free protein complexes within the solvent in the region of the shell $\phi_{\text{fc}} \approx \phi_{\text{bulk}}$. We derive in the Appendix 4 that the probability of an active monomer being bound to a protein complex is given by

$$\Omega \approx \frac{\eta}{1 + \eta},\tag{18}$$

where we introduced the binding parameter $\eta = e^{1+\epsilon} \phi_{\text{bulk}}$ (see Appendix 4 for details).

The equality of the osmotic pressures in the shell region and in the bulk solution $\Pi_{\text{shell}} - \Pi_{\text{bulk}} = 0$, *Equation 56*, can thus be approximated by

$$-\frac{3}{2\sigma^2 \phi_A} + \frac{v_\eta}{2} \phi_A^2 + \frac{w_\eta}{3} \phi_A^3 \approx 0,\tag{19}$$

where we introduced the interfacial area per block $s = S/n$ (whose dimensionless form is written $\sigma = s/a^2$), an effective second virial coefficient

$$v_\eta = \frac{v + (1 + 2\eta)^2}{(1 + \eta)^2},\tag{20}$$

and an effective third virial coefficient

$$w_\eta = \frac{(1 + 2\eta)^3}{(1 + \eta)^3}.\tag{21}$$

It is important to note that in our model the second and third virial coefficients can be tuned by the binding of protein complexes. Both virial coefficients increase with $\eta$, which can be controlled in two ways, by increasing the binding energy $\epsilon$ or the concentration of the protein complexes in the bulk solution, $\phi_{\text{bulk}}$. The solvent regime experienced by the active chromatin brush is then regulated by

the binding of protein complexes and it can range from poor solvent, $v_\eta < 0$, to good solvent, $v_\eta > 0$, passing through the theta solvent regime when $v_\eta \approx 0$. In general, the different solvent regimes correspond to different scaling properties of the radius of gyration of single homopolymers in solution as a function of the number of monomers in the polymer. The different scaling regimes $R_G \sim N^{3/5}$, $R_G \sim N^{1/2}$, and $R_G \sim N^{1/3}$ correspond to good, theta, and poor solvent regimes, respectively (*Rubinstein and Colby, 2003*). However in the case of polymer brushes, the scaling laws also depend on the grafting density of such polymers. In our model, only the active monomers experience different solvent regimes depending on whether the protein complexes bind to them and their surface density is fixed from a balance of the different free energy contributions in the system. This self-assembling situation differs from a polymer brush in which the surface density is considered to be fixed.

We discuss below the different parameter regimes and find approximate solutions for the volume fraction of the active chromatin brush $\phi_A$ and its corresponding dimensionless extension, $\lambda_A = L_A/a$, in each of these regimes.

## Weak binding modifies the length of the collapsed active chromatin blocks

In the weak binding limit, the self-attraction of active chromatin blocks is reduced by the weak binding of protein complexes to the active monomers, but not enough to change the solvent of the active chromatin brush from a poor one to a good one. In this weak binding regime, characterized by $\eta < |v|^{1/2}/2 - 1/2$, the effective second virial coefficient is balanced by the effective third virial coefficient (the second and third terms in *Equation 19*, respectively), which leads to a volume fraction of active monomers

$$\phi_A^c \sim \frac{|v_\eta|}{w_\eta},$$ (22)

and to an extension of the active chromatin brush

$$\lambda_A^c \sim \frac{N_A w_\eta}{v_\eta \, \sigma}.$$ (23)

These solutions show that the volume fraction of monomers in the shell is reduced by the presence of protein complexes bound to active chromatin compared to a system without protein complexes. In the poor solvent regime, the extension of the active brush scales linearly with the number of monomers (*Halperin, 1988*), similarly to a brush in good solvent (*Alexander, 1977*; *de Gennes, 1980*). However, there is a crucial difference between the two, which is that they scale differently with the interfacial area per block, $\sigma$. This different scaling as a function of the interfacial area per block then results (at the end of the calculation) in different scaling behaviors of the active chromatin brush height, the thickness of the core, and the interfacial area per block as a function of the number of active monomers per block.

## Strong binding of protein complexes swells the active chromatin brush

When the protein complexes strongly bind to the active monomers, characterized by $\eta > (v/2)^{1/2} - 1/2$, or $v_\eta > 0$, the effective solvent quality for the active blocks changes from poor to good. In that case, the relevant terms in the osmotic pressure balance are the ones corresponding to the stretching of the active brush and the effective second virial coefficient from *Equation 19*. Balancing those two terms, we find

$$\phi_A^e \sim \frac{1}{v_\eta^{1/3} \sigma^{2/3}},$$ (24)

which predicts an extension of the active brush:

$$\lambda_A^e \sim \frac{N_A}{v_\eta^{1/3} \sigma^{1/3}}.$$ (25)

This is equivalent to the scaling of the height of a polymer brush in a good solvent in a mean field approximation *Alexander, 1977*; *de Gennes, 1980*, with the modification coming from the fact that here the second virial coefficient is controlled by the binding of protein complexes. We have then

shown that protein complexes can modify the effective solvent quality experienced by the active chromatin blocks depending on the extent of binding.

## Theta solvent conditions for the active chromatin blocks

We have shown that depending on the binding strength or concentration of the protein complexes, the active chromatin brush can experience poor or good solvent conditions. The transition between the two occurs at the so-called theta pointde (*de Gennes, 1979*), where the second virial coefficient in the system vanishes (the effective second virial coefficient in our case) and the active blocks behave as if they were ideal polymer chains. In this regime, we find that the volume fraction of active monomers is

$$\phi_A^\Theta \sim \frac{1}{w_\eta^{1/4}\sigma^{1/2}} \,, \tag{26}$$

and that the extension of the brush is

$$\lambda_A^\Theta \sim \frac{N_A w_\eta^{1/4}}{\sigma^{1/2}} \,. \tag{27}$$

If we compare the three regimes in which the brush experiences poor, good, and theta solvent conditions, we find that in all of them, the brush height scales linearly with the number of active monomers in a block, but with different scalings as a function of the surface per block, $\sigma$. Unlike tethered brushes where the density at the surface is fixed, in this system the interfacial density of monomers is adjusted by the osmotic pressure equality (equivalent to minimization of the free energy of the reservoir and the core-shell system for constant total volume), which leads to different scaling behaviors as a function of the number of monomers in the active blocks.

## Core size reduction driven by binding of protein complexes

We now explore the effect of the binding of protein complexes on the size of the inactive core in the different solvent regimes. We begin by solving the equation for the osmotic pressure balance that determines the interfacial area $\sigma$ per block, *Equation 57*, and find its values in the poor (collapsed), good (extended), and theta solvent regimes: $\sigma^c$, $\sigma^e$ and $\sigma^\Theta$, respectively. The interfacial area per block in the poor solvent regime is given by

$$\sigma^c \sim \frac{w_\eta^{2/3} N_A^{1/3}}{v_\eta^{2/3}\alpha^{1/3}} \,, \tag{28}$$

where we introduced the dimensionless interfacial tension, $\alpha = \gamma a^2/k_B T$. In the good solvent regime we find:

$$\sigma^e \sim \frac{v_\eta^{2/5} N_A^{3/5}}{\alpha^{3/5}} \,, \tag{29}$$

and finally, in the theta solvent regime the surface area per block is

$$\sigma^\Theta \sim \frac{w_\eta^{1/4} N_A^{1/2}}{\alpha^{1/2}} \,. \tag{30}$$

We find that the surface area per block increases with the protein concentration in bulk solution and the binding energy (lumped into the binding parameter $\eta$) in every regime and that its scaling with the number of monomers in the active blocks is, as expected, most pronounced in the good solvent regime ($\sigma \sim N_A^{3/5}$).

We now combine the results for the surface area per block with the conservation of inactive monomers to find the dimensionless thickness of the core, $\lambda_B = L_B/a$ and the dimensionless extension of the active chromatin brush, $\lambda_A$. In what follows we neglect numerical prefactors - the full expressions are given in Appendix 5. In the poor solvent regime, the thickness of the core, $\lambda_B^c$, and the extension of the active chromatin brush, $\lambda_A^c$, are given by

$$\lambda_B^c \sim \frac{N_B |v_\eta|^{2/3} \alpha^{1/3}}{N_A^{1/3} w_\eta^{2/3} \phi_B}, \quad \lambda_A^c \sim \frac{N_A^{2/3} w_\eta^{1/3} \alpha^{1/3}}{|v_\eta|^{1/3}}. \tag{31}$$

The corresponding values in the good solvent regime for the thickness of the core and for the extension of the active chromatin brush are

$$\lambda_B^e \sim \frac{N_B \alpha^{3/5}}{N_A v_\eta^{2/5} \phi_B}, \quad \lambda_A^e \sim N_A^{4/5} v_\eta^{1/5} \alpha^{1/5}, \tag{32}$$

and in the case of the theta solvent they are

$$\lambda_B^\theta \sim \frac{N_B \alpha^{1/2}}{N_A^{1/2} w_\eta^{1/4} \phi_B}, \quad \lambda_A^\theta \sim N_A^{3/4} w_\eta^{1/8} \alpha^{1/4}. \tag{33}$$

The scaling behaviors shown in *Equations 31–33* show that the presence of protein complexes that can bind to active chromatin regions can induce transitions between these different regimes in which the thickness of the core is smallest in the good solvent regime (proportional to $1/N_A$ as opposed to fractional powers thereof). In every regime the surface area per block, $\sigma$, and the extension of the active brush, $\lambda_A$, are increasing functions of the binding parameter $\eta$. This means that increasing the binding interaction $\epsilon$ or the concentration of the protein complexes in the bulk solution, $\phi_{bulk}$, will always lead to swelling of the active chromatin brush. In contrast, we find that the core thickness is a decreasing function of $\eta$. A summary of all scaling behaviors is presented in *Appendix 5—table 1*. These results show that although the protein complexes do not interact directly with the inactive blocks, they have a significant effect on the characteristic length scale of inactive chromatin cores. This has particular significance for the core-shell structure in spherical geometry (core-shell micelles) as we now discuss.

## Organization of chromatin in core-shell spherical domains

Thus far, we have showed that protein complexes can lead to changes in the quality of the solvent experienced by the active chromatin brush, which in turn lead to different scaling behaviors of the surface area per block and the relative extensions of both the active and inactive blocks as a function of the binding parameter $\eta$ and the number of monomers in the active blocks, $N_A$. We performed the calculations in a layer geometry for simplicity, however, it is known that block copolymer, microphase-separated states, can also form spherical micelles with a core comprising the blocks experiencing the poorer solvent conditions and a corona comprising the other type of blocks (*Halperin, 1987*; *Mayes and Olvera de la Cruz, 1988*; *Halperin, 1991*; *Zhulina et al., 2005*). Moreover, in the experimental data shown in *Figure 1*, the core-shell organization is reminiscent of spherical micelles. The core contains dense chromatin and the outer shell is composed of chromatin coated with RNA Pol II. Because of these reasons, it is then relevant to discuss the case of chromatin organization in core-shell spherical domains. Below, we provide some analytical insight based on our calculations in the layer geometry and complement this with Brownian dynamics simulations showing that indeed, the analytical predictions give us a qualitative understanding of how protein complexes could regulate the microphase separation of active and inactive chromatin organized in spherical domains.

### Analytical predictions for the number of blocks in spherical core-shell chromatin micelles

If instead of considering the layer geometry - which is simple to treat analytically but might not be the global minimum of the free energy - we consider the case of spherical micelles, we can make a simple extrapolation of the results obtained in the layer geometry presented in the previous section as follows: the thickness of the core, $L_B$, corresponds to the radius $r$ of the spherical core and the interfacial area per block $\sigma$ would be, up to numerical coefficients, equivalent to the area spanned by each active block at the interface of the spherical cores and the outer shells. Doing so allows us to use the conservation of B monomers to estimate the number of blocks, $m$, that are incorporated into each core-shell spherical domain in each regime and how these quantities scale with the number of monomers of the active blocks. The monomer conservation of B monomers in each core is expressed by

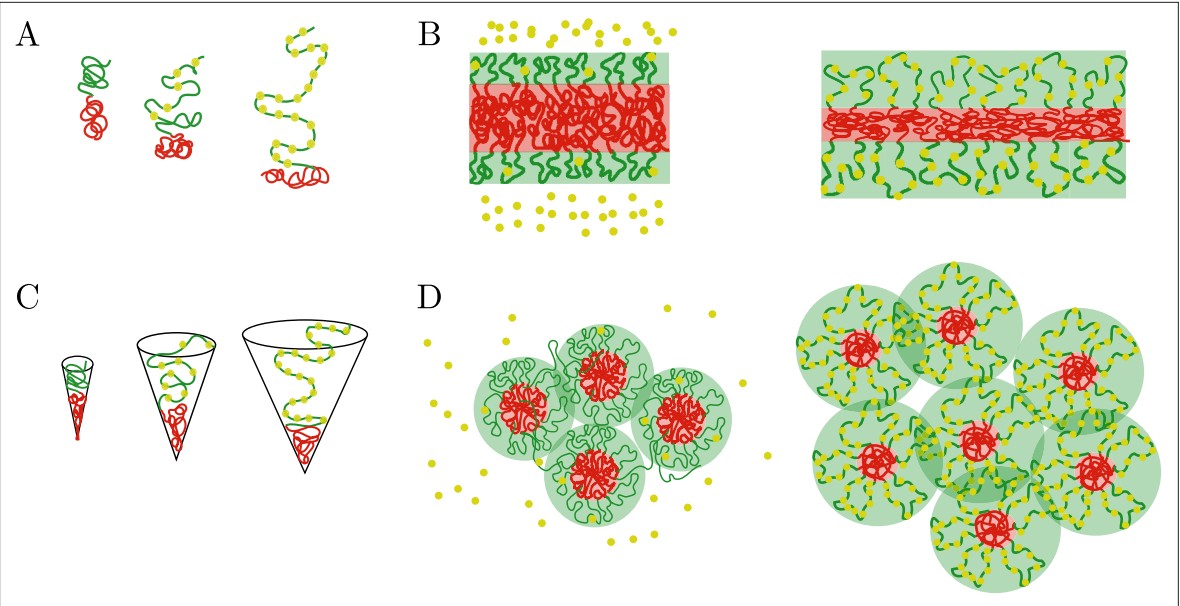

**Figure 4.** Comparison of the chromatin microphase separation by the binding of protein complexes in layer and spherical geometries. (**A**) Single pair of active (green) and inactive blocks (green) in the layer geometry with no (left), few (middle), and many (right) protein complexes bound to a chromatin active block. The swelling of the active block by the binding of protein complexes changes the surface area per block and compresses the inactive blocks. (**B**) Global changes in the microphase-separated layer geometry. Left: The active shell (green) and inactive core (red) remain unchanged in the absence of bound protein complexes (yellow). Right: Binding of protein complexes leads to swelling of the active shell (green region) and compression of the thickness of the core (red region). (**C**) Single pair of active (green) and inactive blocks (green) in the spherical geometry with no (left), few (middle), and many (right) protein complexes bound to the active chromatin block. A preferred radius of curvature is set by the binding of protein complexes, such preferred radius of curvature defines the size of the core. (**D**) Regulation of core number and size by the binding of protein complexes. Left: Weak binding of protein complexes (yellow beads) to active chromatin has little or no effect in the size and number of cores (red area). Right: Strong binding of protein complexes to active chromatin swells the active shells (green area) and increases the number of inactive cores while decreasing their size.

$$\frac{4\pi r^3 \phi_B}{3} = a^3 N_B \, m \, . \tag{34}$$

We now use *Equation 34* to calculate the number of blocks in each core-shell spherical domain in the different regimes. In the poor solvent regime we find:

$$m^c \sim \frac{N_B^2 \alpha |v_\eta|^2}{N_A w_\eta^2 \phi_B^2} \, , \tag{35}$$

in the good solvent regime:

$$m^e \sim \frac{N_B^2 \alpha^{9/5}}{N_A^{9/5} v_\eta^{6/5} \phi_B^2} \, , \tag{36}$$

and in the $\theta$-solvent regime

$$m^\theta \sim \frac{N_B^2 \alpha^{3/2}}{N_A^{3/2} w_\eta^{3/4} \phi_B^2} \, , \tag{37}$$

We can now relate the results for the layer geometry with the spherical core-shell organization. Let us consider a single active block connected to a single inactive block in the layer geometry, as we have shown in the previous sections, increasing the binding strength of the protein complexes to the active chromatin monomers tends to swell the active blocks and concomitantly increase the surface area per block (*Figure 4A*). The swelling of the active chromatin blocks then leads to a compression of the inactive block, which in the global organization of the microphase-separated state translates into a decrease in the thickness of the core and an increase in the interfacial area (*Figure 4B*). If we now consider a single active block connected to a single inactive block in the spherical geometry,

each block spans a conical surface. The solid angle of such cones increases with the binding of protein complexes and we intuitively suggest that just as in the case of the blocks in the layer geometry, the inactive blocks will be more compressed when the binding of the protein complexes to the active block is stronger (*Figure 4C*). If we now focus on the global organization of the microphase-separated state, the number of inactive cores as well as their size is regulated by the binding of protein complexes; the stronger such binding is, more and smaller inactive cores there are (*Figure 4D*). Thus, our minimal model suggests that nuclear size regulation (*Deviri and Safran, 2022*) may have a strong impact in the spatial organization of chromatin in living cells via changes in the concentration of chromatin-binding protein complexes.

In the calculations presented above for the spherical geometry, we have considered a homogeneous monomer concentration in both the core and shell domains. Thus, our calculation excludes the regime in which the active blocks are much larger than the blocks at the core and thus exhibit star polymer behavior (*Daoud and Cotton, 1982*), where the monomer concentration in each 'blob' decreases with its distance from the core. This is however not the focus of our manuscript; instead, we want to highlight the strong effects that selective binding of proteins to active regions of chromatin can have on the global chromatin organization. We show below results of Brownian dynamic simulations for the case $N_A = N_B$ that are in qualitative agreement with our analytical results; such simulations can include the case of active blocks that are much larger than the core which are more difficult to treat analytically.

## Brownian dynamics simulations of chromatin microphase separation regulated by the binding of protein complexes

Our computational model simulates chromatin as a block copolymer comprising both active and inactive blocks. It additionally considers that protein complexes only bind to the active chromatin blocks. Chromatin is considered to be a self-attractive block copolymer (*Maeshima et al., 2016a*; *Gibson et al., 2019*; *Strickfaden et al., 2020*; *Amiad-Pavlov et al., 2021*; *Bajpai et al., 2021*) and is modeled as a bead-spring polymer consisting of $N$ beads connected by $N - 1$ springs (harmonic bonds). These $N$ beads are divided between periodic blocks of active and inactive monomers. Attractions between inactive monomers are taken to be larger than those between active monomers and the cross-interaction of active and inactive monomers. This gives rise to the core-shell structure observed in the experiments, shown in *Figure 1*. In the simulations, when any two beads of the polymer approach each other within a distance $2.5a$ (where $a$ is the diameter of bead), they interact via a truncated Lennard-Jones potential, whose strong repulsive regime also prevents them from overlapping (*Bajpai et al., 2021*). The polymer contains two types of beads: active and inactive, that represent respectively, the active and silent genes in the chromosome. The protein complexes are modeled by additional, single beads that only bind to the active polymer beads; there are no attractive interactions between protein-protein and protein-inactive beads. For simplicity, we assume the size of the protein complex beads to be equal to the size of a chromatin bead, whose diameter is $a$. When a protein bead approaches an active bead within a distance of $1.5a$, they form a bond with probability 1. For other separations of the protein and the active bead, the bond energy increases beyond its minimal value and is modeled by a harmonic potential with bonding strength $K$. Once an active bead and a protein complex bead form a bond, their valencies are saturated and they are not available for bonding. Our model also allows existing bonds to break. When the distance between protein complexes and active chromatin beads exceeds $2.5a$, the bond is broken (there is no longer an attractive force between the beads) and the protein complex is free to diffuse in solution. The chromatin polymer and the protein complexes are confined within a spherical boundary. The simulations were performed using the Brownian dynamics simulation package LAMMPS (*Plimpton, 1995*; *Thompson et al., 2022*), which solves Newton's equations using viscous forces and a Langevin thermostat ensuring an NVT ensemble (see section Computational model for details). Our initial condition has $n = 500$ blocks of the chromatin chain, with each block having 100 beads of active genes ($N_A$) and 100 beads of inactive genes ($N_B$), for a total of $n(N_A + N_B) = 100,000$ beads.

A summary of our numerical results is presented in *Figure 5*, which shows that the increased binding of protein complexes to the active regions swell the active chromatin regions and reduce the inactive core sizes, in agreement with the analytical scaling results presented above. In *Figure 5A*, we show snapshots of simulations for different binding strengths. It is clear from such snapshots that the

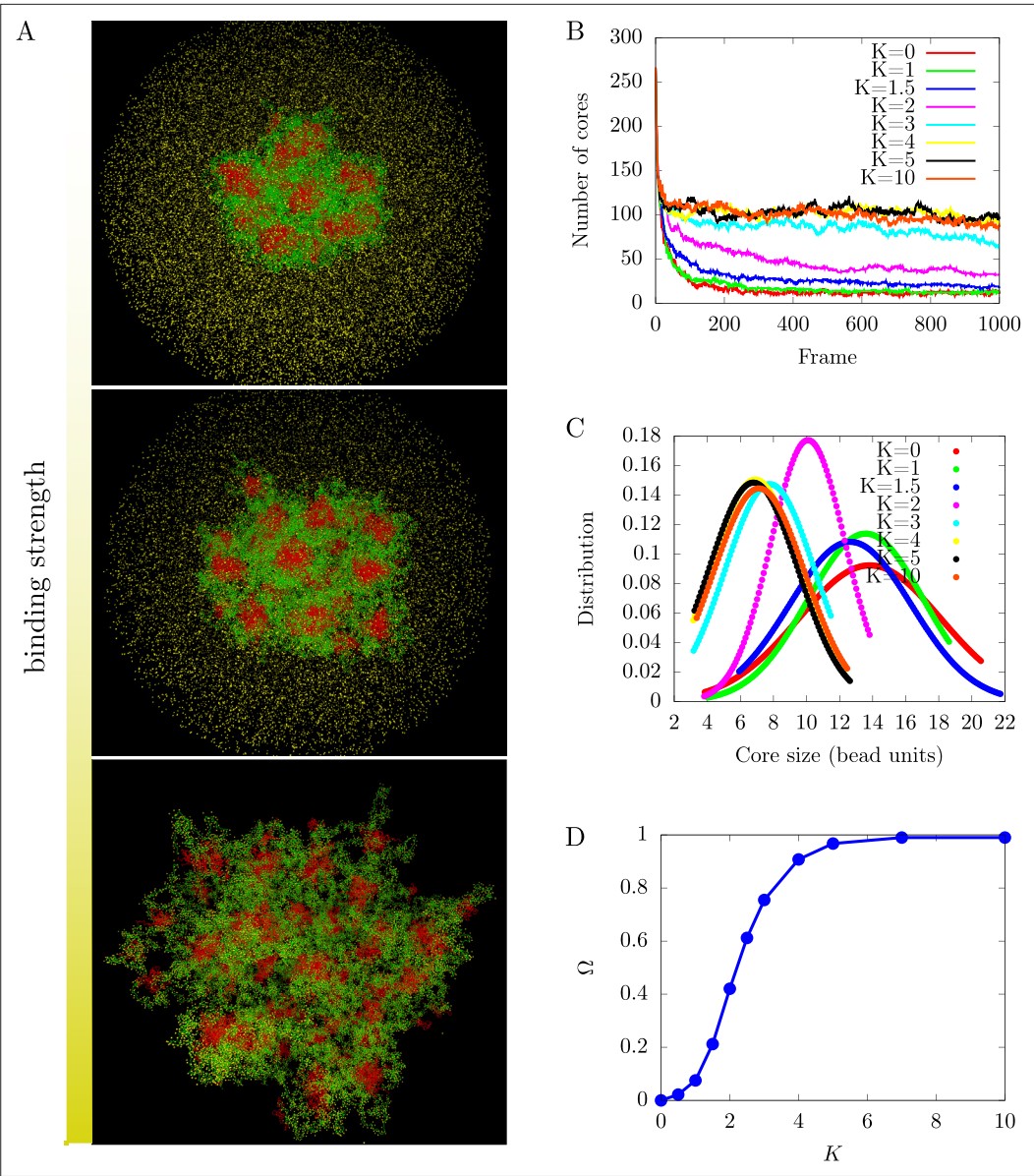

**Figure 5.** Regulation of chromatin microphase separation by binding of protein complexes. (**A**) Simulation snapshots show a core-shell organization of chromatin. The cores (red area) are composed of inactive blocks (red lines) of chromatin, and the shells (green area) are composed of active blocks (green lines) of chromatin. The snapshots from top to bottom correspond to binding of protein complexes (yellow points) to chromatin where deviations of the bond distance from its minimal value are modeled by a harmonic potential with a spring constant $K$. $K = 0$ (no binding, top), $K = 2$ (weak binding, middle), and $K = 10$ (strong binding, bottom). (**B**) Number of cores as a function of the simulation time frame by varying the harmonic potential characterized by $K$ of protein complexes. The number of cores increases with increasing bonding strength until it saturates. (**C**) Core size distributions for different bonding strengths, $K$. (**D**) Fraction of active monomers bound to a protein complex, $\Omega$, as a function of bonding strengths follows a sigmoidal curve.

increase in binding strength leads to more extended configurations of the active chromatin regions, an increase of the number of inactive cores, and a reduction in their size (as required by the conservation of inactive monomers). We also show in *Figure 5B* that the number of cores at steady state, increases as a function of binding strength but then saturates to a value of around 100. The saturation is a simple consequence of the fact that in this saturation limit, all active monomers have a protein complex bound to them. The shell extends no further even if the binding energy or protein concentration is increased and the core size is also saturated at its minimal value. We show the probability

distribution of core sizes measured in bead units in *Figure 5C*. We find that the distributions are slightly broader at low bonding strength and that they transition to narrower distributions for values of $K \gtrsim 3$. We observe three different regimes: (i) fully collapsed, with $K \lesssim 1$, (ii) a transition region from fully collapsed to fully stretched going from $1 \lesssim K \lesssim 3$, and (iii) the fully stretched regime which is given by $3 \lesssim K$. Finally, the fraction of active monomers that have a protein complex bound is shown in *Figure 5D*, where we see a sigmoidal behavior for $\Omega$ as a function of the binding strength, which is characterized by the strength of the harmonic potential between the protein complexes and the active chromatin, $K$. This is what gives rise to the saturation mentioned above. In this saturation limit, for $K \gtrsim 4$, the fraction of active monomers that are bound to a protein complex is approximately 1. The simulation results agree with the predictions of the analytical theory, and confirm that the binding of protein complexes can serve as regulator of the microphase separation of active and inactive chromatin regions.

## Discussion

We have shown theoretically and by computer simulations that protein complexes binding to active regions of chromatin can have a significant effect in the nuclear scale organization of chromatin. This is accomplished via stretching of the active chromatin blocks when they are bound to protein complexes, which leads to compaction of the region in which the inactive blocks are located. This then results, in the case of the spherical core-shell organization, in an increase in the number of cores comprising inactive chromatin regions and a reduction of their size. We used a minimal, generic model of chromatin as a multiblock copolymer to which protein complexes can bind, this in turn modulates the effective solvent quality vis a vis the chromatin. The effect is strongly dependent on the concentration of the protein complexes and their binding energy. Depending on these two parameters, there are two different regimes, weak and strong binding. In each regime we predict different scaling relations for the extension of the active chromatin brushes and core thickness, as well as the global organization of the core-shell assemblies, as a function of the number of monomers in each of these blocks. We predict that the number of micelles and the relative core-shell sizes in living cells, such as the ones shown in *Figure 1*, are not fixed but will change as the RNA Pol II and other chromatin-binding protein concentration are varied. We show a sketch in *Figure 6* exemplifying such change. Moreover, if we contrast our results with experimental observations in which chromatin exhibits several different nuclear types of organization, depending on whether the cell is fixed (*Cremer et al., 2015*), live cultured cells (*Eshghi et al., 2021*), or in intact live cells (*Amiad-Pavlov et al., 2021*), we suggest that changes in the hydration of the nucleus can lead to different local concentrations of chromatin-binding proteins which in turn can regulate the nature of the microphase separation of active and inactive chromatin.

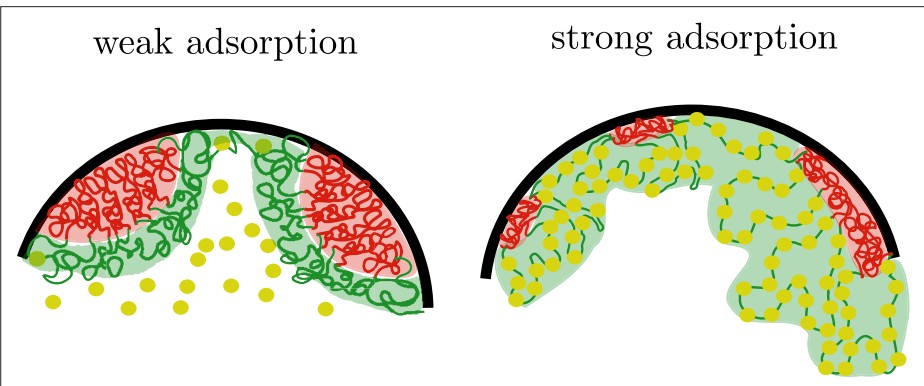

**Figure 6.** Sketch of chromatin microphase separation in confinement. Left: Weak binding of protein complexes (yellow) to active chromatin (green) results in larger inactive chromatin domains (red) that wet the nuclear envelope (black) as spherical caps. Right: Strong binding of protein complexes to the active chromatin leads to smaller inactive chromatin domains and might also result in changes of their shape going from spherical to cylindrical-like domains. It must be noted that the volume of the inactive domains is conserved in the three-dimensional organization.

In this work, we have provided a generic description at the mesoscale, that is independent of the microscopic details that determine chromatin microphase separation. Instead, we suggest a minimal mechanism that can play a role in regulating the microphase separation between active and inactive chromatin regions. Although this has been previously observed experimentally (*Hilbert et al., 2021*), a model such as ours has not been proposed. With our generic model we provide a complementary interpretation of previously published experiments on the effect of transcription in chromatin microphase separation. In *Hilbert et al., 2021*, the authors show that transcription organizes chromatin in microphase-separated states, with RNA Pol II connecting an RNA-rich phase with a chromatin-rich phase. In that study, drug treatments that lead to unbinding of RNA Pol II from chromatin result in an *increase* in the sizes of high-density chromatin domains; notably, those domains are fewer in number than in the wild type and they do not seem to fully coarsen. This is in agreement with our minimal theory, where in the absence of protein complexes, the dense cores become larger but not fully coalesce due to the selectivity of the solvent. Another treatment they used stopped transcription but did not lead to the unbinding of RNA Pol II from chromatin; in those cells, they still observed a similar distribution of dense chromatin domains, pointing to the fact that chromatin remains in its microphase-separated state despite the fact that there are no active processes regulating it. The authors of that study relate Pol II to its 'amphiphilic' role in connecting the DNA-rich region which is transcribed to produce nascent RNA. We suggest a complementary scenario in which binding of RNA Pol II to chromatin is the important factor that regulates the sizes and number of dense (inactive) chromatin domains as we discuss above. It has also been shown recently in *Leidescher et al., 2022*, that long transcription loops span very large regions of the nucleus and are coated by RNA Pol II. The authors rationalized the extension of the loops as a consequence of the large size of the nascent RNAs that are produced. This provides an interesting direction in which our work could be extended, in which one introduces not only the binding of protein complexes but also the transcription products that are carried by such protein complexes.

In addition to the observations provided in this article for chromatin being in a microphase-separated state, previous work (*Popken et al., 2014*; *Cremer et al., 2015*) has focused on the different chromatin compartments in terms of their density, classifying them into active and inactive nuclear compartments, and showing clear boundaries between them. More recently (*Gelléri et al., 2022*), those authors showed evidence of dense chromatin cores surrounded by extended chromatin regions, which seems to be in agreement with our proposed model of chromatin microphase separation.

Our equilibrium approach can be extended by considering non-equilibrium effects such as acetylation and methylation of the histone tails of nucleosomes, which would lead to different sizes of active and inactive chromatin blocks. The simplest non-equilibrium scenario, would be one where the fraction of protein complexes bound to chromatin, $\Omega$, is fixed by kinetic binding and unbinding rates, instead of detailed balance (and the Boltzmann factor) as in equilibrium. In this scenario, the results obtained for the equilibrium system will hold as long as the reaction kinetics are much faster than the equilibration of the core-shell system. The core-shell organization would then be controlled by the steady-state bound fraction $\Omega$ set by the binding and unbinding rates instead of by *Equation 18*. Another extension to the model would be to investigate the possible interplay arising from protein complexes that actively extend the active chromatin blocks and by doing so create a positive feedback for histone tail acetylation by enzymes, which in turn, would create more binding sites for the protein complexes. Finally, it would be relevant to asses the role of nuclear confinement in chromatin microphase separation by including the possibility of the association of chromatin with the nuclear envelope via its lamina-associated domains (*van Steensel and Belmont, 2017*), as shown in *Figure 6*. This would further regulate the organization of chromatin and might give rise to different microphase-separated states. The binding of chromatin to the nuclear lamina might further regulate the size and shape of the inactive domains and have an additional effect on the spatiotemporal organization of chromatin and DNA transcription.

## Model
### Brownian dynamics simulations

In order to model the chromatin system, we use the following potential energies. The stretching energy of the chromatin chain is provided by harmonic springs that connect adjacent beads and is given by

$$U_{\text{stretching}} = k_{\text{spring}} \sum_{i=1}^{N-1} (|\mathbf{r}_i - \mathbf{r}_{i+1}| - a)^2, \tag{38}$$

where $\mathbf{r}_i$ and $\mathbf{r}_{i+1}$ are the position vectors of $i^{\text{th}}$ and $(i+1)^{\text{th}}$ beads, respectively. Here, $a$ is the equilibrium bond length, and $k_{\text{spring}}$ represents the spring constant. In our simulations, spring constant is set to a large value, $k_{\text{spring}} = 100 \, k_{\text{B}} T / a^2$, to ensure that the bonds are stable and do not fluctuate a lot.

Non-bonded interactions between chromatin-chromatin and chromatin-protein beads are modeled using the standard Lennard-Jones potentials (LJ),

$$U_{\text{LJ}} = 4\epsilon_{\alpha\beta} \sum_{i<j} \left[ \left( \frac{a}{|\mathbf{r}_i - \mathbf{r}_j|} \right)^{12} - \left( \frac{a}{|\mathbf{r}_i - \mathbf{r}_j|} \right)^6 \right], \tag{39}$$

for all $|\mathbf{r}_i - \mathbf{r}_j| < r_c$, where $r_c$ refers to a cutoff distance beyond which LJ interaction is set to zero. Here, $\epsilon_{\alpha\beta}$ is the strength of the LJ potential. For the attractive interactions between chromatin-chromatin and chromatin-protein beads, $r_c = 2.5a$ is used as the distance cutoff for the LJ potential. In homopolymers, the polymer chain starts to collapse at $\epsilon_{\alpha\beta} = 0.3 \, k_{\text{B}} T$ (**Bajpai and Safran, 2022**). Chromatin chain beads are divided into active and inactive types based on their attractive strength. Active chromatin beads are set at minimum attractive strength, and LJ strength between any two active beads is taken to be $\epsilon_{AA} = 0.3 \, k_{\text{B}} T$. The LJ strength of $\epsilon_{BB} = 0.5 \, k_{\text{B}} T$ is set between two inactive beads, and the strength of $\epsilon_{AB} = 0.3 \, k_{\text{B}} T$ between active beads and inactive beads. The attraction of chromatin-protein beads is modeled using the same LJ potential and $\epsilon_{Bp} = 0.4 \, k_{\text{B}} T$ is taken as LJ strength between active chromatin and protein beads. There are no attractive interactions between protein-protein beads or inactive chromatin and protein beads (only excluded volume interactions), so the LJ potential is truncated at the distance where repulsive and attractive forces are equal which gives $r_c = 2^{1/6} a$ and $\epsilon_{pp} = \epsilon_{Ap} = 1 \, k_{\text{B}} T$. Beads of polymer chains and protein are confined within a sphere of radius $R_c$. The interaction between the beads and the sphere surface is repulsive, given by

$$U_{\text{confine}} = 4\epsilon_{\text{confine}} \sum_{i=1}^{N} \left[ \left( \frac{a}{R_c - |\mathbf{r}_i|} \right)^{12} - \left( \frac{a}{R_c - |\mathbf{r}_i|} \right)^6 \right], \tag{40}$$

for all $(R_c - |\mathbf{r}_i|) < 2^{1/6} a$ and $\epsilon_{\text{confine}} = 1 \, k_{\text{B}} T$. To estimate $R_c$, we define the parameter $\phi$ as the volume fraction of chain within the sphere volume

$$\phi = \frac{\text{Volume of chromatin beads}}{\text{Volume of confinement}} = \frac{N \times \frac{4}{3} \pi (a/2)^3}{\frac{4}{3} \pi R_c^3}, \tag{41}$$

where $N$ is the total number of beads in chromatin chain and $N = 100,000$ beads are taken in our model. In order to observe the effect of Pol II complex proteins on the swelling of the active blocks of the chromatin, we minimize the effect of the nucleus by assuming that the chromatin volume fraction within the nucleus is 1% ($\phi = 0.01$). We find the confinement radius $R_c = 108a$ by plugging the value of $\phi$ into **Equation 41**.

A major feature of the chromatin-protein complex interaction in our model is the bonding between active beads and protein complex beads. This depends on the availability of these proteins and the nature of their binding to the active chromatin. For the simplicity, we consider a total of $N_p = 50,000$ protein beads, which is equal to the total number of active chromatin beads so that the volume fraction of protein beads within the nucleus is 0.005; each active bead can bind to at most one protein complex bead. The active bead and protein complex bead each have a valence of 1. The formation of harmonic bonds between active and protein complexes occurs when they are within a defined interaction cutoff distance ($r_{\text{bond}} = 1.5a$). The spring constant ($K$) for the changes in the bond distance

is varied from 0 to $10\,k_BT/a^2$. Bonding interactions have an equilibrium distance of $r_0 = a$, which means a bond, once formed, remains stable at that distance. In addition, dynamics bonds can break (potential goes to zero) when the distance between the active beads and protein complex beads exceeds the cutoff $r_{break}$. We set the cutoff value $r_{break}$ to equal the LJ cutoff $r_c$ at which the LJ interaction also vanishes; therefore $r_{break} = 2.5a$. The potential energy of bonds creating and breaking ($U_{bond}$) is calculated by using the Monte Carlo method in LAMMPS software (*Plimpton, 1995*; *Thompson et al., 2022*) and written by

$$U_{bond} = -U_0(r_{bond}, r_0) + K\sum_{i=1}^{N}\sum_{j=1}^{N_p}(|\mathbf{r}_i - \mathbf{r_p}_j| - r_0)^2, \tag{42}$$

for all $|\mathbf{r}_i - \mathbf{r_p}_j| < r_{break}$, where $r_{break}$ refers to the cutoff distance beyond which dynamic bond interaction ($U_{bond}$) is set to zero. Here, $\mathbf{r_p}$ is the position of the protein beads, and the energy function $U_0(r_{bond}, r_0)$ represents the bond formation energy. The total potential energy of chromatin system is given by

$$U_{tot} = U_{stretching} + U_{LJ} + U_{confine} + U_{bond} \tag{43}$$

To simulate the chromatin system, we performed Brownian dynamics simulations using the LAMMPS molecular dynamics package (*Plimpton, 1995*; *Thompson et al., 2022*). The simulations were conducted with reduced units ($a = 1$ and $k_B = 1$) at a constant temperature $T = 1.0$ with a damping coefficient of $10.0\tau$, where $\tau$ is the time unit. In all simulations, a time-step $\Delta t = 0.01\tau$ was used. Simulations were run for $10^7$ time-steps that is much longer than the time it takes for the system to reach a steady state with constant mean energy and radius of gyration.

## Acknowledgements

OAA acknowledges funding by the Armando and Maria Jinich fellowship for Mexican Citizens. OAA and GB acknowledge the Feinberg Graduate School. DL and TV acknowledge the Israel Science Foundation (ISF) grant # 750/17. DL, TV, and SAS acknowledge the Tandem Call Weizmann - PIC3i Curie. SAS acknowledges funding by the Volkswagen Foundation and the Weizmann-Caltech fund. SAS is also grateful to the Perlman family foundation for their historic support. The authors are thankful to Amit Kumar and Dan Deviri for helpful discussions.

## Additional information

### Funding

| Funder | Grant reference number | Author |
| --- | --- | --- |
| Volkswagen Foundation | 98 197 | Samuel Safran |
| Feinberg Graduate School, Weizmann Institute of Science | | Gaurav Bajpai Omar Adame-Arana |
| Israel Science Foundation | 750/17 | Dana Lorber Talila Volk |

The funders had no role in study design, data collection and interpretation, or the decision to submit the work for publication.

### Author contributions

Omar Adame-Arana, Conceptualization, Formal analysis, Supervision, Validation, Investigation, Visualization, Methodology, Writing - original draft, Project administration, Writing - review and editing; Gaurav Bajpai, Conceptualization, Data curation, Software, Formal analysis, Investigation, Visualization, Methodology, Writing - original draft, Writing - review and editing; Dana Lorber, Conceptualization, Resources, Validation, Investigation, Visualization, Writing - review and editing; Talila Volk, Supervision, Funding acquisition, Writing - review and editing; Samuel Safran, Conceptualization,

Supervision, Funding acquisition, Investigation, Writing - original draft, Project administration, Writing - review and editing

### Author ORCIDs

Omar Adame-Arana ⬤ http://orcid.org/0000-0001-5828-0584
Gaurav Bajpai ⬤ http://orcid.org/0000-0003-3875-4599
Dana Lorber ⬤ http://orcid.org/0000-0002-0635-8703
Talila Volk ⬤ http://orcid.org/0000-0002-3800-2621
Samuel Safran ⬤ http://orcid.org/0000-0002-0798-1492

### Decision letter and Author response

Decision letter https://doi.org/10.7554/eLife.82983.sa1
Author response https://doi.org/10.7554/eLife.82983.sa2

## Additional files

### Supplementary files

• MDAR checklist

### Data availability

All data generated or analyzed during this study are included in the manuscript and supporting files. The LAMMPS code and source data files of simulations are available on Zenodo using the following link: https://zenodo.org/record/7977176.

The following dataset was generated:

| Author(s) | Year | Dataset title | Dataset URL | Database and Identifier |
| --- | --- | --- | --- | --- |
| Adame-Arana O, Bajpai G, Lorber D, Volk T, Safran SA | 2023 | Regulation of chromatin microphase separation by adsorbed protein complexes | https://doi.org/10.5281/zenodo.7977176 | Zenodo, 10.5281/zenodo.7977176 |

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

# Appendix 1

## RNA Pol II distribution in live salivary gland epithelium

In this appendix, we show a representative RNA Pol II distribution in live *Drosophila* salivary gland epithelium. We observe that RNA Pol II is distributed mostly in the surrounding shell of dense chromatin domains.

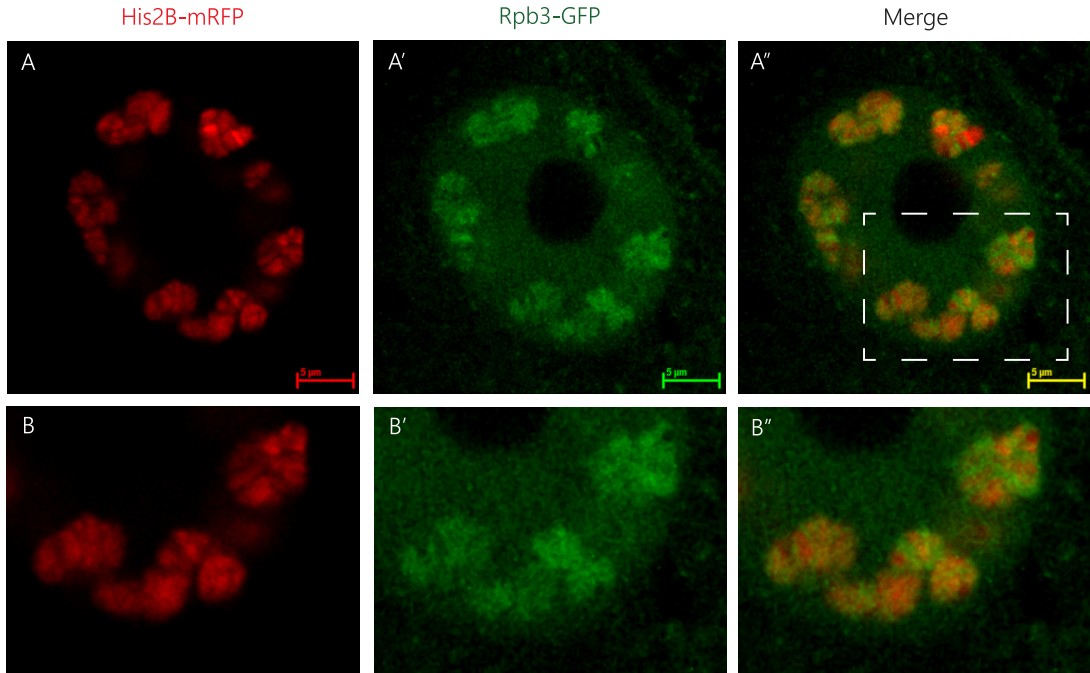

**Appendix 1—figure 1.** Distribution of RNA Pol II in live *Drosophila* salivary gland epithelium. The salivary gland in intact live larvae expressing His2B under endogenous promoter (Red), and Rpb3-GFP under Mef2Gal4 driver imaged under the microscope. (**A**) His2B-RFP representing H2B-associated chromatin, (**A'**) Rpb3-GFP representing Pol II-associated chromatin, and **A"** their merged image. (**B–B"**) are corresponding enlargements of the rectangle in **A"**.

## Appendix 2

### Derivation of the equilibrium conditions the core-shell chromatin system in a solution of protein complexes

The equilibrium conditions shown in the main text can be derived starting from the following unconstrained, grand potential - equivalent to minimizing the free energy with various conservation constraints:

$$\mathcal{L} = F_{\text{core}} + F_{\text{shell}} + F_{\text{bulk}} + \mu \left( N_{\text{T}} - N_{\text{fc}} - N_{\text{d}} - N_{\text{bulk}} \right)$$
$$- \Pi \left( V_{\text{T}} - V_{\text{core}} - 2L_{\text{A}}S - V_{\text{bulk}} \right) , \tag{44}$$

where $\mu$ is a Lagrange multiplier used to enforce the conservation of protein complexes in the system, $N_{\text{T}}$ is the total number of protein complexes, $\Pi$ is a Lagrange multiplier that enforces volume conservation, $V_{\text{T}}$ is the total volume. $V_{\text{core}}$ is the volume occupied by the core, which we approximate to be constant; this is appropriate in the limit where the interactions that determine the core concentration are much larger than the energies of the brush, so that the core monomer concentration is constant. We now minimize the unconstrained free energy, $\mathcal{L}$, with respect to the following six variables: the number of free protein complexes in the shell, $N_{\text{fc}}$, the number of protein complexes bound to the active chromatin brush, $N_{\text{b}}$, the number of protein complexes in the bulk solution $N_{\text{bulk}}$, the thickness of the shell region, $L_{\text{A}}$, with respect to the surface area of the interface between core and shell, $S$, and the volume of the bulk protein solution, $V_{\text{bulk}}$. The minimization leads to the following conditions:

$$\frac{\partial F_{\text{shell}}}{\partial N_{\text{fc}}} - \mu = 0 , \tag{45}$$

$$\frac{\partial F_{\text{shell}}}{\partial N_{\text{d}}} - \mu = 0 , \tag{46}$$

$$\frac{\partial F_{\text{bulk}}}{\partial N_{\text{bulk}}} - \mu = 0 , \tag{47}$$

$$\frac{\partial F_{\text{shell}}}{\partial L_{\text{A}}} - 2S\,\Pi = 0 , \tag{48}$$

$$\frac{\partial F_{\text{core}}}{\partial S} + \frac{\partial F_{\text{shell}}}{\partial S} + 2L_{\text{A}}\Pi = 0 , \tag{49}$$

$$\frac{\partial F_{\text{bulk}}}{\partial V_{\text{bulk}}} + \Pi = 0 . \tag{50}$$

Combining *Equations 45–50*, we obtain *Equations 12, 15, and 17*, which are the equations that we solve for the equilibrium state of the system. There are six conditions coming from the minimization of the unconstrained grand potential which determine the six (unknown) degrees of freedom listed above. These conditions fix the equalities of the exchange chemical potentials of the protein complexes (in the bulk, bound to the A blocks, and in the solution within the shell) as well as the equality of the osmotic pressure of the bulk solution and the brush (shell region) (*Equation 8* and *Equation 9*). There are two ways to change the volume: (i) by changing the interfacial of the core-shell and (ii) by changing the extent of the shell due to stretching of the A blocks. These give two osmotic pressure conditions. Since we work in the limit in which the bulk solution is much larger than the core and the active chromatin brush regions, we consider that the exchange chemical potential and osmotic pressures in the bulk solution can be varied independently.

## Appendix 3

### Equality of exchange chemical potentials and osmotic pressures in the core-shell system

We now use the composition variables introduced in the main text, namely, the volume fraction of free protein complexes, $\phi_{\mathrm{fc}}$, the fraction of active monomers with a protein complex bound to them, $\Omega$, the volume fraction of active monomers in the active brush, $\phi_{\mathrm{A}}$, and the surface area per block $s = S/n$, to express the exchange chemical potentials and osmotic pressures in the system. These are given by

$$\mu_{\mathrm{fc}} = k_{\mathrm{B}}T(\log \phi_{\mathrm{fc}} - \log(1 - \phi_{\mathrm{fc}} - \phi_{\mathrm{A}}(1 + \Omega))), \tag{51}$$

$$\mu_{\mathrm{d}} = k_{\mathrm{B}}T \left( \log \Omega - \log(1 - \phi_{\mathrm{fc}} - \phi_{\mathrm{A}}(1 - \Omega)) \right.$$
$$\left. - \log(1 - \Omega) - 1 - \epsilon + v\phi_{\mathrm{A}}(1 - \Omega) \right), \tag{52}$$

$$\Pi_{\mathrm{shell}} = \frac{k_{\mathrm{B}}T}{a^3} \left( \frac{-3a^4}{2s^2 \phi_{\mathrm{A}}} - \phi_{\mathrm{A}}(1 + \Omega + v\phi_{\mathrm{A}}(1 - \Omega)^2) \right.$$
$$\left. - 2\log(1 - \phi_{\mathrm{fc}} - \phi_{\mathrm{A}}(1 + \Omega)) \right). \tag{53}$$

The other two expressions that are needed to find solutions to the equilibrium conditions are the exchange chemical potential of protein complexes in the bulk, $\mu_{\mathrm{bulk}}$, and osmotic pressure in the bulk, $\Pi_{\mathrm{bulk}}$, which are given in the main text in *Equation 8* and *Equation 9*, respectively.

## Appendix 4

### Solutions to the equilibrium conditions

Here, we explicitly write the four equilibrium conditions used in the main text. The condition of equal exchange chemical potentials of the protein complexes in the bulk and in the shell, *Equation 11*, is given by

$$\log(\phi_{\text{fc}}) - \log(\phi_{\text{bulk}}) - \log(1 - \phi_{\text{fc}} - \phi_{\text{A}}(1 + \Omega)) = 0 \,, \tag{54}$$

the equality of exchange chemical potentials of free and bound protein complexes in the shell can be expressed as

$$1 + \epsilon + v\phi_{\text{A}}(1 - \Omega) + \log \phi_{\text{fc}} + \log\left(\frac{1 - \Omega}{\Omega}\right) = 0, \tag{55}$$

the balance of osmotic pressures in the bulk solution and the shell is given by

$$\frac{3a^4}{2s^2\phi_{\text{A}}} + \phi_{\text{bulk}} + \phi_{\text{A}}\left(1 + \Omega + v\phi_{\text{A}}(1 - \Omega)^2\right)$$
$$\log(1 - \phi_{\text{fc}} - \phi_{\text{A}}(1 + \Omega)) = 0 \,, \tag{56}$$

and finally, the second osmotic pressure condition, equivalent to defining the interfacial tension with the free energy per block surface, *Equation 17*, can be expressed as

$$\gamma - \frac{3k_{\text{B}}Ta^4 N_{\text{A}}}{2s^3\phi_{\text{A}}} = 0 \,. \tag{57}$$

### Shell region

In this section, we focus on approximate analytical expressions for the volume fraction of the protein complexes in the shell region and for the fraction of active monomers bound to a protein complexes. We obtain these approximate analytical expressions based on the physical conditions discussed above: *Equations 54, 55*. We begin by solving *Equation 54* and find

$$\phi_{\text{fc}} = \frac{\phi_{\text{bulk}}(1 - \phi_{\text{A}}(1 + \Omega))}{1 + \phi_{\text{bulk}}} \approx \phi_{\text{bulk}} \,, \tag{58}$$

where we assumed, $\phi_{\text{bulk}} \ll 1$, and neglected terms of the form $\phi_{\text{bulk}}\phi_{\text{A}}$, on the right-hand side. Solving *Equation 55* by substituting $\phi_{\text{fc}} \approx \phi_{\text{bulk}}$, we find that the fraction of active monomers that have a protein complex bound to them is

$$\Omega \approx \frac{e^{1+\epsilon}\phi_{\text{bulk}}}{1 + e^{1+\epsilon}\phi_{\text{bulk}}} \,, \tag{59}$$

In the main text we analyze the first osmotic pressure balance, *Equation 56*, by substituting the values for the volume fraction of free protein complexes in the shell region, $\phi_{\text{fc}}$, and for the fraction of active monomers bound to a protein complex, $\Omega$.

## Appendix 5

### Length scales of the microphase-separated state in the layer geometry

Here, we write the full expressions for the z-direction extent of the shell and the core in the regimes corresponding to different effective solvent conditions of the shell. We note that changes in the extent of the shell and the interfacial area per block also affect the extent of the core.

**Appendix 5—table 1.** Characteristics of the flat inactive core and active shell for different regimes effective solvent regime.

The effective second and third virial coefficients are $v_\eta = (v + (1 + 2\eta)^2)/(1 + \eta)^2$ and $w_\eta = (1 + 2\eta)^3/(1 + \eta)^3$, respectively. In the table we use the following dimensionless quantities: the extension of the active brush, $\lambda_A = L_A/a$, the extension of the inactive blocks, $\lambda_B = L_B/a$, the surface area per block $\sigma = s/a^2$, and the interfacial tension, $\alpha = \gamma a^2/(k_B T)$.

| Poor solvent | Good solvent | Theta solvent |
|---|---|---|
| $\lambda_A^c \approx \dfrac{1}{2^{1/3} 3^{2/3}} \dfrac{N_A^{2/3} w_\eta^{1/3} \alpha^{1/3}}{\lvert v_\eta \rvert^{1/3}}$ | $\lambda_A^e \approx \dfrac{1}{2^{4/5} 3^{2/5}} N_A^{4/5} v_\eta^{1/5} \alpha^{1/5}$ | $\lambda_A^\Theta \approx \dfrac{1}{2^{5/8} 3^{1/2}} N_A^{3/4} w_\eta^{1/8} \alpha^{1/4}$ |
| $\lambda_B^c \approx \dfrac{3^{1/3}}{2^{4/3}} \dfrac{N_B \lvert v_\eta \rvert^{2/3} \alpha^{1/3}}{N_A^{1/3} w_\eta^{2/3} \phi_B}$ | $\lambda_B^e \approx \dfrac{1}{2^{2/5} 3^{1/5}} \dfrac{N_B \alpha^{3/5}}{N_A^{3/5} v_\eta^{2/5} \phi_B}$ | $\lambda_B^\Theta \approx \dfrac{1}{2^{3/4}} \dfrac{N_B \alpha^{1/2}}{N_A^{1/2} w_\eta^{1/4} \phi_B}$ |
| $\sigma^c \approx \dfrac{2^{1/3}}{3^{1/3}} \dfrac{w_\eta^{2/3} N_A^{1/3}}{\lvert v_\eta \rvert^{2/3} \alpha^{1/3}}$ | $\sigma^e \approx \dfrac{3^{1/5}}{2^{3/5}} \dfrac{N_A^{3/5} v_\eta^{2/5}}{\alpha^{3/5}}$ | $\sigma^\Theta \approx \dfrac{1}{2^{1/4}} \dfrac{N_A^{1/2} w_\eta^{1/4}}{\alpha^{1/2}}$ |

