## [Editor Report]

This fundamental work substantially advances our understanding of polymer physics underpinnings of genome folding, organization, and regulation. The conclusions are supported by both convincing computer simulations and analytical theory. The work will be of significant interest to the genome folding community.

---

## [Decision Letter]

**Decision letter after peer review:**

Thank you for submitting your article "Regulation of chromatin microphase separation by binding of protein complexes" for consideration by *eLife*. Your article has been reviewed by 3 peer reviewers, one of whom is a member of our Board of Reviewing Editors, and the evaluation has been overseen by Aleksandra Walczak as the Senior Editor. The following individual involved in the review of your submission has agreed to reveal their identity: Alexander Y. Grosberg (Reviewer #2).

Essential revisions:

1) Explain to the reader the non-trivial and counter-intuitive character of their central assumption – that RNAPs, although attracted to DNA, make solvent effectively better, not worse.

*Reviewer #1 (Recommendations for the authors):*

1. The authors should additional data for the existence of the micro-domains studied in this work and for the dependence of their sizes on at least one parameter that also enters the theoretical analysis.

2. Improve the explanation of physics concepts for biologists. This does not mean repeating passages from textbooks, but it should have the aim of providing intuition of various physical expressions to biologists.

*Reviewer #2 (Recommendations for the authors):*

As a polymer physicist, first and foremost, I encourage authors to clarify the confusion which may or may not be purely terminological, but definitely precludes a complete understanding of the work. The issue is most obvious around Equation 20, where authors explicitly claim that the negative second virial coefficient corresponds to the good solvent regime, while the positive one to the poor solvent. This is the exact opposite of the commonly accepted nomenclature, as any polymer physics textbook can confirm (e.g., Rubinstein and Colby, or Grosberg and Khokhlov, etc).

My second recommendation is to take into account the fact that presently a lot of work is being done on the statistical physics of the so-called active systems. Due to this, when authors mention that some parts of the chromatin are active – the implication is that they are consuming energy by some mechanism, which would then preclude the use of equilibrium statistical mechanics. I encourage authors to clean up the terminology appropriately.

My third recommendation is to pay significantly more attention to the discussion of applicability conditions of the (many!) approximations adopted in the work. This includes the assumption that core density is fixed and is not affected by proteins as well as the assumption that the shell phase consists of loops (starting from a core and coming back to the same core) rather than numerous strands connecting different cores.

My fourth recommendation is to consider framing the work in terms of the strong segregation limit of block copolymers (properly referring to the appropriate papers, starting, e.g., from Semenov, Macromolecules 1993, 26, 24, 6617) instead of referring to the seminal papers treating the weak segregation limit. I think that the work really considers structures with sharp interfaces between microphases.

My fifth recommendation for the authors is to emphasize the nature of their assumption about the role of proteins in changing interactions between parts of the chromatin fiber. The usual situation with DNA-binding proteins is that they are positively charged. As such, they can usually serve as cross-links or condensing agents for DNA, as evidenced by the example of DNA behavior in presence of even three- or four-valent ions (such as spermine and spermidine). Authors assume that RNA polymerase complexes do something almost entirely opposite to the fiber, they attach to one place on the fiber and somehow generate the repulsion from another section (i.e, provide, in common nomenclature, a positive contribution to the second virial coefficient). This is a highly unusual assumption that is at the heart of the suggested model and should be properly stated and explained.

*Reviewer #3 (Recommendations for the authors):*

The paper is written very clearly. But please explain in more detail two points:

1) The factor (1-\Omega)^2^ in Equation 5: Do I understand right that this factor means that you assume that monomers that are bound by proteins are not attracted to each other anymore? This, I guess, might be the crucial place where you change the effective solvent quality of the active blocks through the binding of the proteins. If so, this should be explicitly stated close to the formula and a rationalization of this assumption should be provided.

2) On page 13 you write: "When any two beads of the polymer approach each other within a distance 2.5 \σ a […], they are attracted by a truncated LJ potential." For me this sounded like all monomers interact with the same strength which then made the micellar structure observed on top of Fig, 5A puzzling. Only after reading the appendix, it became clear to me that inactive beads attract each other stronger. This (namely that inactive monomers have a stronger attraction to each other) should simply be mentioned already in the main text.

---

## [Author Response]

Essential revisions:Reviewer #1 (Recommendations for the authors):1. The authors should additional data for the existence of the micro-domains studied in this work and for the dependence of their sizes on at least one parameter that also enters the theoretical analysis.

The image is a representative image and we have more evidence of RNA Pol II organizing at the periphery of dense chromatin regions. We include a new figure in Appendix 1 showing that the coreshell organization is also observed in salivary glands. There is published evidence in Hilbert, et al. Nat. Comm. 2021, showing that dissociation of protein complexes (RNA Pol II and transcription factors) from the chromatin fiber leads to the coarsening of chromatin domains, consistent with our model predictions. We refer to this in the discussion and write the following in line 567:

“In Hilbert et al. (2021), the authors show that transcription organizes chromatin in microphase separated states, with RNA Pol II connecting an RNA-rich phase with a chromatin-rich phase. In that study, drug treatments that lead to unbinding of RNA Pol II from chromatin result in an increase in the sizes of high density chromatin domains; notably, those domains are fewer in number than in the wild type and they do not seem to fully coarsen. This is in agreement with our minimal theory, where in the absence of protein complexes, the dense cores become larger but not fully coalesce due to the selectivity of the solvent. Another treatment they used, stopped transcription but did not lead to the unbinding of RNA Pol II from chromatin; in those cells, they still observed a similar distribution of dense chromatin domains, pointing to the fact that chromatin remains in its microphase separated state despite the fact that there are no active processes regulating it.”

2. Improve the explanation of physics concepts for biologists. This does not mean repeating passages from textbooks, but it should have the aim of providing intuition of various physical expressions to biologists.

We thank the reviewer for this suggestion. We have now provided other explanatory remarks in the revised version of our manuscript, as follows:

Lines 252 to 274

“…where we introduced the interaction between active monomers, *v*, which we consider to be attractive (in the absence of protein binding), that is *v* < 0. One core assumption in our model is that the presence of bound protein complexes reduce the effective attraction between active monomers. The rationale is based in the following: The protein complexes that bind to active chromatin tend to be relatively large, (e.g., RNA Pol II pre-initiation complex), in which a myriad of proteins assemble jointly with RNA Pol II in order to initiate transcription, which can lead to steric repulsion due to the large macromolecular volume of such complexes Hahn (2004). Moreover, while RNA Pol II interacts with the DNA in the chromatin fiber via positively charged patches at its surface, most of its surface is negatively charged Cramer et al. (2001). Hence, we assume that when RNA Pol II is bound to the chromatin fiber, those Pol II regions that are negatively charged would effectively provide a repulsive force within the chromatin fibers. Another possible explanation is that in the vicinity of active monomers, the transcription complex carries mRNA, which is known to segregate from chromatin Hilbert et al. (2021). While a more thorough analysis of the molecular interactions would be interesting, it is out of the scope of our paper that uses a coarse-grained, polymer physics approach. This approach also allows our model to be predictive as to the physical organization and growth of the domains, independent of those molecular details that are as yet unknown.

The free energy we consider in Equation 5 is appropriate in the limit that the surface density of the contacts between active and inactive blocks always remains relatively high; therefore active blocks are heavily influenced by the presence of neighboring blocks. In this limit, the system behaves like a polymer brush. In the other limit that the surface density of contacts of the active and inactive brushes is sufficiently low, the appropriate expression would be that of isolated mushrooms in which each block is independent of each other and their description is equivalent to that of single polymers which obey simple scaling laws, depending on the solvent quality Szleifer (1996).”

Lines 330 to 339

“In general, the different solvent regimes correspond to different scaling properties of the radius of gyration of single homopolymers in solution as a function of the number of monomers in the polymer. The different scaling regimes R_G_ ∼*N*
^3/5^, R_G_ ∼*N*
^1/2^, and R_G_ ∼*N*
^1/3^, correspond to good, theta, and poor solvent regimes, respectively Rubinstein et al. (2003). However in the case of polymer brushes, the scaling laws also depend on the grafting density of such polymers. In our model, only the active monomers experience different solvent regimes depending on whether the protein complexes bind to them and their surface density is fixed from a balance of the different free energy contributions in the system. This self-assembling situation differs from a polymer brush in which the surface density is considered to be fixed.”

Referring to a theta solvent, we added some text between lines 378 to 380

“…at the so called theta point de Gennes (1979), where the second virial coefficient in the system vanishes (the effective second virial coefficient in our case) and the active blocks behave as if they were ideal polymer chains.”

Reviewer #2 (Recommendations for the authors):As a polymer physicist, first and foremost, I encourage authors to clarify the confusion which may or may not be purely terminological, but definitely precludes a complete understanding of the work. The issue is most obvious around Equation 20, where authors explicitly claim that the negative second virial coefficient corresponds to the good solvent regime, while the positive one to the poor solvent. This is the exact opposite of the commonly accepted nomenclature, as any polymer physics textbook can confirm (e.g., Rubinstein and Colby, or Grosberg and Khokhlov, etc).

We thank the referee for pointing out the inconsistency with the polymer physics literature. This inconsistency stems from the fact that we explicitly wrote a minus sign in the definition of the interaction between monomers belonging to what we called active blocks in order to emphasize their attractive nature. In the revised version, we rewrote the equations in a manner that is consistent with the established literature. Namely we defined negative, zero and positive virial coefficients corresponding to poor, theta and good solvent conditions.

To keep our nomenclature in line with the polymer physics literature, we made changes in Equations 5, 19, 22, 24, 25, 29, 31, 32, 35 and 36.

My second recommendation is to take into account the fact that presently a lot of work is being done on the statistical physics of the so-called active systems. Due to this, when authors mention that some parts of the chromatin are active – the implication is that they are consuming energy by some mechanism, which would then preclude the use of equilibrium statistical mechanics. I encourage authors to clean up the terminology appropriately.

The biology literature refers to active and inactive chromatin, and generally denote euchromatin (less condensed) and heterochromatin (more condensed) regions. The less condensed regions correspond to parts of the chromatin that can be transcribed whereas those that are more condensed are less or not transcribed at all. Thus, active and inactive in this literature refer to transcription and not necessarily to self-propelled systems etc. that are the concern of “active matter physics”. It is true, however, that energy consumption via ATP may be involved in transcription; but this is outside the level of description of our polymeric model. To prevent confusion, we have now added the following clarification when the terms are introduced in the manuscript to avoid confusion. In line 68 we write:

“Hereafter we use the name active for regions of chromatin which are found in regions of lower chromatin density that would commonly be associated with genes that may undergo transcription, It is important to note that this use of active is different from activity as used in the physics literature (e.g., active swimmers) on non-equilibrium forces in systems with internal energy sources.”

Our use of the terms active and inactive is consistent with the nomenclature used by different authors in the biological and biophysical literature who also denote active and inactive compartments, for example, Cremer, et al., BioEssays 42, 2020 and Hildebrand and Dekker, Trends Biochem Sci. 2020 May; 45(5): 385–396.

My third recommendation is to pay significantly more attention to the discussion of applicability conditions of the (many!) approximations adopted in the work. This includes the assumption that core density is fixed and is not affected by proteins as well as the assumption that the shell phase consists of loops (starting from a core and coming back to the same core) rather than numerous strands connecting different cores.

In the revised paper we point out the approximations more explicitly. For example, the approximation of protein complexes being unable to penetrate the dense chromatin cores due to its high density, is inspired by the experimental observations of the lack of penetration of RNA Pol II to the dense chromatin cores as shown in Figure 1. There is further evidence showing that large particles cannot diffuse within inactive chromatin domains, shown by Gelléri, et al., bioRxiv 2022.03.23.485308, 2022. In line 207 we included the following text:

“Our assumption that protein complexes only bind to active monomers stems from the observed lack of penetration of RNA Pol II into the dense chromatin cores as shown in Figure 1. Additionally, there is further evidence showing that large particles cannot diffuse into inactive chromatin domains Gelléri et al. (2022).”

We also state more clearly that our approximation of a fixed concentration of the core which contains the inactive chromatin blocks, is appropriate when the interactions between heterochromatic domains is the largest energy scale in the system, compared to the protein binding, polymer entropy etc. This is also related to a finding in Gelléri, et al., bioRxiv 2022.03.23.485308, 2022, where the chromatin density in heterochromatic domains is much higher than that of chromatin found in active compartments (transcriptionally active domains). We write in line 226:

“Such an approximation is motivated by the fact that dense chromatin domains are known to have much higher concentrations that chromatin found in transcriptionally active compartments Gelléri et al. (2022).”

Regarding considering the loop conformations instead of bridging between different micelles, the main reason to consider such a scenario is to simplify the calculations; we do not know the exact conditions in the nucleoplasm but we expect the qualitative trends of our model to apply independent of the as yet unknown details of the looping.

My fourth recommendation is to consider framing the work in terms of the strong segregation limit of block copolymers (properly referring to the appropriate papers, starting, e.g., from Semenov, Macromolecules 1993, 26, 24, 6617) instead of referring to the seminal papers treating the weak segregation limit. I think that the work really considers structures with sharp interfaces between microphases.

We thank the reviewer for referring us to the seminal work by Semenov. We now cite it appropriately and mention that our model considers the strong segregation limit with sharp interfaces. In line 201 we write:

“Our model considers microphase separation in the strong segregation limit (Semenov (1985), Ohta and Kawasaki (1986)) with sharp interfaces, the strong incompatibility arises here from the strong selectivity of the solvent, in which inactive blocks experience very strong compaction, reminiscent of the much higher density of DNA in inactive compartments.”

My fifth recommendation for the authors is to emphasize the nature of their assumption about the role of proteins in changing interactions between parts of the chromatin fiber. The usual situation with DNA-binding proteins is that they are positively charged. As such, they can usually serve as cross-links or condensing agents for DNA, as evidenced by the example of DNA behavior in presence of even three- or four-valent ions (such as spermine and spermidine). Authors assume that RNA polymerase complexes do something almost entirely opposite to the fiber, they attach to one place on the fiber and somehow generate the repulsion from another section (i.e, provide, in common nomenclature, a positive contribution to the second virial coefficient). This is a highly unusual assumption that is at the heart of the suggested model and should be properly stated and explained.

We thank the reviewer for highlighting the fact that the interactions that we describe are different in essence than that of cross-bridging molecules such as spermine and spermidine. One reason for this is that chromatin consists of DNA and positively charged histones (with hydrophobic tails) and in the cell, with additional chromatin-binding proteins. Chromatin is known to condense in aqueous solvent (see Gibson, et al., Cell. 2019, 179) and has been shown (see Amiad-Pavlov, et al., Science Advances, 2021) to phase separate from the nucleoplasm. This means that water is a poor solvent for chromatin – in contrast to DNA. In our model, the binding of protein complexes to the chromatin disrupts the self-attraction of the chromatin and effectively results in good solvent conditions for those parts of the chain to which the protein complexes have bound. The less condensed nature of transcriptionally active chromatin (to which Pol-II binds) allows us to treat that part of the chromatin as being in good solvent conditions. The treatment of the electrostatics is more complex than for DNA and multivalent proteins since the histone proteins, histone tails, and other associated proteins must be taken into account. We therefore coarse grain all of these effects by distinguishing only between the poor solvent conditions of the transcriptionally inactive chromatin (to which the Pol-II does not bind) and the relatively good solvent conditions of transcriptionally active chromatin to which Pol-II binds. In addition, the main difference between the two molecules, spermine and spermidine and RNA Pol II, among others, is that the surface of RNA Pol II is mainly negatively charged, but with some patches of positive charges that allow for the specific association of RNA Pol II with DNA (see Cramer, et al., Science, 2001). Our model seems to be closely related to the observations made in Hilbert, et al. Nat. Comm. 2021, in which the association of RNA Pol II (and very likely other factors involved in transcription) prevent coarsening of high density chromatin domains even in the absence of transcription. We include a clarification of this effect in the description of our model in line 253:

“One core assumption in our model is that the presence of bound protein complexes reduce the effective attraction between active monomers. The rationale is based in the following: The protein complexes that bind to active chromatin tend to be relatively large, (e.g., RNA Pol II pre-initiation complex), in which a myriad of proteins assemble jointly with RNA Pol II in order to initiate transcription, which can lead to steric repulsion due to the large macromolecular volume of such complexes Hahn (2004). Moreover, while RNA Pol II interacts with the DNA in the chromatin fiber via positively charged patches at its surface, most of its surface is negatively charged Cramer et al. (2001). Hence, we assume that when RNA Pol II is bound to the chromatin fiber, those Pol II regions that are negatively charged would effectively provide a repulsive force within the chromatin fibers. Another possible explanation is that in the vicinity of active monomers, the transcription complex carries mRNA, which is known to segregate from chromatin Hilbert et al. (2021). While a more thorough analysis of the molecular interactions would be interesting, it is out of the scope of our paper that uses a coarse-grained, polymer physics approach. This approach also allows our model to be predictive as to the physical organization and growth of the domains, independent of those molecular details that are as yet unknown.”

Reviewer #3 (Recommendations for the authors):The paper is written very clearly. But please explain in more detail two points:1) The factor (1-\Omega)^2^ in Equation 5: Do I understand right that this factor means that you assume that monomers that are bound by proteins are not attracted to each other anymore? This, I guess, might be the crucial place where you change the effective solvent quality of the active blocks through the binding of the proteins. If so, this should be explicitly stated close to the formula and a rationalization of this assumption should be provided.

We are grateful that the referee appreciated a crucial feature of our model: the chromatin monomers, in the absence of protein, are self-attractive (are in poor solvent) and the protein binding disrupts this selfattraction, resulting in relatively good solvent conditions for the parts of the chain bound to the Pol II proteins. We expanded our description of the model and make this assumption more explicit in line 252 we write:

“…where we introduced the interaction between active monomers, *v*, which we consider (in the absence of protein binding) to be attractive, that is *v* < 0. One core assumption in our model is that the presence of bound protein complexes reduce the effective attraction between active monomers. The rationale is based in the following: The protein complexes that bind to active chromatin tend to be relatively large, (e.g., RNA Pol II pre-initiation complex), in which a myriad of proteins assemble jointly with RNA Pol II in order to initiate transcription, which can lead to steric repulsion due to the large macromolecular volume of such complexes Hahn (2004). Moreover, while RNA Pol II interacts with the DNA in the chromatin fiber via positively charged patches at its surface, most of its surface is negatively charged Cramer et al. (2001). Hence, we assume that when RNA Pol II is bound to the chromatin fiber, those Pol II regions that are negatively charged would effectively provide a repulsive force within the chromatin fibers. Another possible explanation is that in the vicinity of active monomers, the transcription complex carries mRNA, which is known to segregate from chromatin Hilbert et al. (2021). While a more thorough analysis of the molecular interactions would be interesting, it is out of the scope of our paper that uses a coarse-grained, polymer physics approach. This approach also allows our model to be predictive as to the physical organization and growth of the domains, independent of those molecular details that are as yet unknown.”

2) On page 13 you write: "When any two beads of the polymer approach each other within a distance 2.5 \σ a […], they are attracted by a truncated LJ potential." For me this sounded like all monomers interact with the same strength which then made the micellar structure observed on top of Fig, 5A puzzling. Only after reading the appendix, it became clear to me that inactive beads attract each other stronger. This (namely that inactive monomers have a stronger attraction to each other) should simply be mentioned already in the main text.

We thank the reviewer for this comment, and agree that we should have definitely mentioned the stronger attraction of inactive beads in the main text already. We have added the following text in line 192 when introducing the model:

“Attractions between inactive monomers are taken to be larger than those between active monomers and the cross interaction of active and inactive monomers”.

List of changes marked in red in the revised manuscript.

In line 17 we added “and transcription factors”

In line 68 we added “Hereafter we use the name active for regions of chromatin which are found in regions of lower chromatin density that would commonly be associated with genes that may undergo transcription, It is important to note that this use of active is different from activity as used in the physics literature (e.g., active swimmers) on non-equilibrium forces in systems with internal energy sources.”

In the caption of Figure 1. we added “Representative image”.

In line 166 we added “In Appendix 1 we show another representative confocal image of the distribution of chromatin and RNA Pol II in a salivary gland of an intact *Drosophila* larva”

In line 201 we added “Our model considers microphase separation in the strong segregation limit Semenov (1985); Ohta and Kawasaki (1986) with sharp interfaces, the strong incompatibility arises here from the strong selectivity of the solvent, in which inactive blocks experience very strong compaction, reminiscent of the much higher density of DNA in inactive compartments.”

In line 207 we added “Our assumption that protein complexes only bind to active monomers stems from the observed lack of penetration of RNA Pol II into the dense chromatin cores as shown in Figure 1. Additionally, there is further evidence showing that large particles cannot diffuse into in active chromatin domains Gelléri et al. (2022).”

In line 226 we added “Such an approximation is motivated by the fact that dense chromatin domains are known to have much higher concentrations that chromatin found in transcriptionally active compartments Gelléri et al. (2022).”

In line 252 we added “where we introduced the interaction between active monomers, *v*, which we consider (in the absence of protein binding) to be attractive, that is *v* < 0. One core assumption in our model is that the presence of bound protein complexes reduce the effective attraction between active monomers. The rationale is based in the following: The protein complexes that bind to active chromatin tend to be relatively large, (e.g., RNA Pol II pre-initiation complex), in which a myriad of proteins assemble jointly with RNA Pol II in order to initiate transcription, which can lead to steric repulsion due to the large macromolecular volume of such complexes Hahn (2004). Moreover, while RNA Pol II interacts with the DNA in the chromatin fiber via positively charged patches at its surface, most of its surface is negatively charged Cramer et al. (2001). Hence, we assume that when RNA Pol II is bound to the chromatin fiber, those Pol II regions that are negatively charged would effectively provide a repulsive force within the chromatin fibers. Another possible explanation is that in the vicinity of active monomers, the transcription complex carries mRNA, which is known to segregate from chromatin Hilbert et al. (2021). While a more thorough analysis of the molecular interactions would be interesting, it is out of the scope of our paper that uses a coarse-grained, polymer physics approach. This approach also allows our model to be predictive as to the physical organization and growth of the domains, independent of those molecular details that are as yet unknown.

The free energy we consider in Equation 5 is appropriate in the limit that the surface density of the contacts between active and inactive blocks always remains relatively high; therefore active blocks are heavily influenced by the presence of neighboring blocks. In this limit, the system behaves like a polymer brush. In the other limit that the surface density of contacts of the active and inactive brushes is sufficiently low, the appropriate expression would be that of isolated mushrooms in which each block is independent of each other and their description is equivalent to that of single polymers which obey simple scaling laws, depending on the solvent quality Szleifer (1996)*”*

In line 331 we added “In general, the different solvent regimes correspond to different scaling properties of the radius of gyration of single homopolymers in solution as a function of the number of monomers in the polymer. The different scaling regimes *R*_G_ ∼*N*
^3/5^, *R*_G_ ∼*N*
^1/2^, and *R*_G_ ∼*N*
^1∕3^, correspond to good, theta, and poor solvent regimes, respectively Rubinstein et al. (2003). However in the case of polymer brushes, the scaling laws also depend on the grafting density of such polymers. In our model, only the active monomers experience different solvent regimes depending on whether the protein complexes bind to them and their surface density is fixed from a balance of the different free energy contributions in the system. This self-assembling situation differs from a polymer brush in which the surface density is considered to be fixed.”

In line 380 we added “and the active blocks behave as if they were ideal polymer chains.”

In line 490 we added “block copolymer”

In line 493 we added “These *N* beads are divided between periodic blocks of active and inactive monomers. Attractions between inactive monomers are taken to be larger than those between active monomers and the cross interaction of active and inactive monomers. This gives rise to the core-shell structure observed in the experiments, shown in Figure 1.”

In line 497 we added “interact”.

In lines 570 to 576 we added “drug treatments that lead to unbinding of RNA Pol II from chromatin result in an increase in the sizes of high density chromatin domains; notably, those domains are fewer in number than in the wild type and they do not seem to fully coarsen. This is in agreement with our minimal theory, where in the absence of protein complexes, the dense cores become larger but not fully coalesce due to the selectivity of the solvent. Another treatment used, stopped transcription but did not lead to the unbinding of RNA Pol II from chromatin, in those cells, they still observed a similar distribution of …”.

We modified Equations 5, 19, 22, 24, 25, 29, 31, 32, 35 and 36 and changed the inequalities in lines 330, 331, 349, 364,

We added Appendix I with the title RNA-Poll II distribution in live *Drosophila* salivary gland epithelium.

We made small modifications to the table appearing in Appendix 5, to keep the consistency with the virial coefficients definitions.